# Optimal Rates for Nonparametric Density Estimation under Communication Constraints

**Jayadev Acharya**
Cornell University
Ithaca, USA
acharya@cornell.edu

**Clément L. Canonne**
University of Sydney
Sydney, Australia
clement.canonne@sydney.edu.au

**Aditya Vikram Singh**
Indian Institute of Science
Bangalore, India
adityavs@iisc.ac.in

**Himanshu Tyagi**
Indian Institute of Science
Bangalore, India
htyagi@iisc.ac.in

## Abstract

We consider density estimation for Besov spaces when the estimator is restricted to use only a limited number of bits about each sample. We provide a noninteractive adaptive estimator which exploits the sparsity of wavelet bases, along with a simulate-and-infer technique from parametric estimation under communication constraints. We show that our estimator is nearly rate-optimal by deriving minmax lower bounds that hold even when interactive protocols are allowed. Interestingly, while our wavelet-based estimator is almost rate-optimal for Sobolev spaces as well, it is unclear whether the standard Fourier basis, which arise naturally for those spaces, can be used to achieve the same performance.

## 1 Introduction

Estimating distributions from samples is a fundamental statistical task. Modern applications, such as those arising in federated learning or the Internet of Things (IoT), often limit access to the true data samples. One common limitation in large scale distributed systems is communication constraints, which require that each data sample must be compressed to a small number of bits.

Most prior work on communication-constrained estimation has focused on parametric problems such as Gaussian mean estimation and discrete distribution estimation. In this work, we study nonparametric density estimation under communication constraints where independent samples from an unknown distribution (whose density $f$ lies in a function class) are distributed across players (one sample per player), and each player can only send $\ell$ bits about its sample to a central referee; the referee outputs an estimate of $f$ based on these $\ell$-bit messages. This problem has been considered before by [5] for densities in Hölder class $\mathcal{H}_L^s([0,1])$ (functions supported on $[0,1]$ satisfying $|f(x) - f(y)| \le L|x-y|^s$ for every $x, y \in [0,1]$), which is a relatively simple class for which the normalized histogram with uniform bins is known to be optimal in the centralized setting. This suggests a natural method for distributed setting – quantize each data sample into uniform bins and use the optimal estimator for distributed discrete distribution estimation. Indeed, [5] shows that this is optimal for distributed estimation of densities from the Hölder class under communication constraints. However, this simple estimator does not seem to extend to the richer Sobolev class and the most general Besov classes. In particular, the following question is largely open:

*How to quantize samples to estimate densities from Besov classes under communication constraints?*

35th Conference on Neural Information Processing Systems (NeurIPS 2021).

We resolve this question both when the density belongs to a Besov class with known parameters (*nonadaptive* setting), and when the density belongs to a Besov class where only upper and lower bounds on parameters are known (*adaptive* setting). Specifically, our proposed estimators exploit the sparsity of wavelet basis for the Besov class, and use vector quantization followed by the distributed simulation technique introduced in [3] for distributed parametric estimation. We also establish lower bounds that prove the optimality of our estimators (up to logarithmic terms in the adaptive setting).

## 1.1 Problem setup

$X_1, \ldots, X_n$ are independent samples from an unknown distribution with density $f$ supported on $\mathcal{X} := [0, 1]$ and belonging to the Besov space $\mathcal{B}(p, q, s)$ (see Section 2 for details). There are $n$ distributed users (*players*), with player-$i$ having access to sample $X_i$. Each player can only transmit $\ell$ bits about its sample to a central server (*referee*) whose goal, upon observing $\ell$-bit messages from $n$ players, is to estimate $f$. We consider an *interactive* setting, where the current player observes the messages from previous players and can use them to design its message.[1] That is, in round $i$, player-$i$ chooses a communication-constrained channel (randomized mapping) $W_i \colon \mathcal{X} \to \{0, 1\}^\ell$ as a function of prior messages $Y_1, \ldots, Y_{i-1}$ and randomness $U$ available to all players; it then passes $X_i$ through $W_i$ to generate $Y_i \in \{0, 1\}^\ell$. The referee observes the messages $Y_1, \ldots, Y_n$ and outputs an estimate $\hat{f}$ of $f$. We term such an $\hat{f}$ an $(n, \ell)$-*estimate*; let $\mathcal{E}_{n,\ell}$ denote the set of all $(n, \ell)$-estimates. Our goal is to design estimators that achieve the minmax expected $\mathcal{L}_r$ loss defined, for $r \geq 1$, by

$$\mathcal{L}_r^*(n, \ell, p, q, s) := \inf_{\hat{f} \in \mathcal{E}_{n,\ell}} \sup_{f \in \mathcal{B}(p,q,s)} \mathbb{E}_f\left[\left\|\hat{f} - f\right\|_r^r\right]. \tag{1}$$

For upper bounds on $\mathcal{L}_r^*(n, \ell, p, q, s)$, we consider algorithms that use the more restricted *noninteractive* protocols, where the channel $W_i$ of player-$i$ is not allowed to depend on the messages $Y_1, \ldots, Y_{i-1}$ or on the common randomness $U$, but it may depend on the private randomness $U_i$ available at player-$i$, where $U_1, \ldots, U_n$ are independent of each other and jointly of $U$. Noninteractive protocols are easier to implement and result in much simpler engineering for the distributed system.

## 1.2 Our results and techniques

Our first result is an information-theoretic lower bound on $\mathcal{L}_r^*$.

**Theorem 1.1.** *For any $p, q, s, r$, there exist constants $C = C(p, q, s, r) > 0, \alpha = \alpha(p, q, s, r) > 0$ such that*

$$\mathcal{L}_r^*(n, \ell, p, q, s) \geq C \cdot \begin{cases} \max\{n^{-\frac{rs}{2s+1}}, \left(n2^\ell\right)^{-\frac{rs}{2s+2}}\}, & r \leq (s+1)p, \\ \max\{n^{-\frac{rs}{2s+1}}, \left(n2^\ell\right)^{-\frac{r(s-1/p+1/r)}{2(s-1/p)+2}} \left(\log n2^\ell\right)^{-\alpha}\}, & r \in ((s+1)p, (2s+1)p), \\ \max\{\left(\frac{n}{\log n}\right)^{-\frac{r(s-1/p+1/r)}{2(s-1/p)+1}}, \left(n2^\ell\right)^{-\frac{r(s-1/p+1/r)}{2(s-1/p)+2}} \left(\log n2^\ell\right)^{-\alpha}\}, & r \geq (2s+1)p. \end{cases}$$

We emphasize that this lower bound even applies to *interactive* protocols as defined in Section 1.1. When the parameters $p, q, s$ of the Besov space are known, we design a *noninteractive* estimator that achieves the optimal rate when $r \leq p$.

**Theorem 1.2.** *For any $r \geq 1$ and $p, q, s$ with $r \leq p$, there exists a constant $C = C(p, q, s, r)$ and an $(n, \ell)$-estimate $\hat{f}$ formed using a noninteractive protocol such that*

$$\sup_{f \in \mathcal{B}(p,q,s)} \mathbb{E}_f\left[\left\|\hat{f} - f\right\|_r^r\right] \leq C \max\{n^{-\frac{rs}{2s+1}}, \left(n2^\ell\right)^{-\frac{rs}{2s+2}}\}.$$

We finally design an *adaptive*, *noninteractive* estimator that only requires bounds on $s$, and no further knowledge of $p$ and $q$. Moreover, this estimator achieves (up to logarithmic factors) the optimal rate for all parameter values.

**Theorem 1.3.** *For any $N \in \mathbb{N}$, $r \geq 1$, and $p, q, s$ with $1/p < s < N$, there exist constants $C = C(p, q, s, r), \alpha = \alpha(p, q, s, r)$ and an $(n, \ell)$-estimate $\hat{f}$ formed using a noninteractive protocol*

---

[1]Since our lower bounds rely on general results in [2], we borrow the notation from that paper.

*such that*

$$\sup_{f \in \mathcal{B}(p,q,s)} \mathbb{E}_f\left[\left\|\hat{f} - f\right\|_r^r\right] \leq C \log^\alpha n \cdot \begin{cases} \max\{n^{-\frac{rs}{2s+1}}, \left(n2^\ell\right)^{-\frac{rs}{2s+2}}\}, & r \leq (s+1)p, \\ \max\{n^{-\frac{rs}{2s+1}}, \left(n2^\ell\right)^{-\frac{r(s-1/p+1/r)}{2(s-1/p)+2}}\}, & r \in ((s+1)p, (2s+1)p), \\ \max\{n^{-\frac{r(s-1/p+1/r)}{2(s-1/p)+1}}, \left(n2^\ell\right)^{-\frac{r(s-1/p+1/r)}{2(s-1/p)+2}}\}, & r \geq (2s+1)p, \end{cases}$$

*where the protocol only requires knowledge of N (an upper bound on s).*

In summary, for all $r \geq 1$, the minmax $\mathcal{L}_r$ loss of estimating $\mathcal{B}(p,q,s)$ (up to logarithmic factors) is

$$\mathcal{L}_r^*(n,\ell,p,q,s) \asymp \begin{cases} \max\{n^{-\frac{rs}{2s+1}}, \left(n2^\ell\right)^{-\frac{rs}{2s+2}}\} & r \leq (s+1)p, \\ \max\{n^{-\frac{rs}{2s+1}}, \left(n2^\ell\right)^{-\frac{r(s-1/p+1/r)}{2(s-1/p)+2}}\} & r \in ((s+1)p, (2s+1)p), \quad (2) \\ \max\{n^{-\frac{r(s-1/p+1/r)}{2(s-1/p)+1}}, \left(n2^\ell\right)^{-\frac{r(s-1/p+1/r)}{2(s-1/p)+2}}\} & r \geq (2s+1)p. \end{cases}$$

It it worth noting that the first term of the maximum in all cases is the standard, unconstrained nonparametric rate (*cf.* [10], or the discussion below), while the second term reflects the convergence slowdown due to the communication constraints. The effect of communication constraints disappears when $\ell$ is sufficiently large. In particular, we get back the centralized rates when $\ell$ satisfies

$$\ell \geq \begin{cases} \left(\frac{1}{2s+1}\right) \log n, & \text{if } r \leq (s+1)p, \\ \left(\frac{2s(1-1/r)}{(2s+1)(s-1/p+1/r)} - \frac{1}{2s+1}\right) \log n, & \text{if } r \in ((s+1)p, (2s+1)p), \\ \left(\frac{1}{2(s-1/p)+1}\right) \log n, & \text{if } r \geq (2s+1)p. \end{cases}$$

For the standard case of $\mathcal{L}_2$ loss, with, say $p \geq 2$, the minmax rate becomes the more interpretable quantity

$$\mathcal{L}_2^*(n,\ell,p,q,s) \asymp \max\{n^{-\frac{2s}{2s+1}}, \left(n2^\ell\right)^{-\frac{2s}{2s+2}}\},$$

where we see that the $\ell$-bit communication constraint reduces the exponent of the convergence rate from $\frac{2s}{2s+1}$ to $\frac{2s}{2s+2}$. Note that the difference becomes less perceptible as $\ell$ grows, or $s$ tends to $\infty$. Finally, from (2) we observe qualitative changes at $r = (s+1)p$ and $r = (2s+1)p$, where the rate exponent changes slope. This phenomenon, whose analogue is observed in the unconstrained setting [10] as well as under local privacy constraints [7], is sometimes referred to as an *elbow effect*.

**Quantize, simulate and infer.** A conceptually simple technique for distributed estimation under communication constraints ("simulate-and-infer") was proposed in [3], which uses communication to *simulate* samples from the unknown distribution, and provides an optimal rate estimator for discrete distribution estimation under communication constraints. A natural approach for nonparametric estimation would be to quantize the samples to the available number of bits ($\ell$) and use this quantized sample to estimate the distribution. However, it is unclear if this approach gives optimal rates. Instead, in our approach, we quantize without inducing any "loss of information." Specifically, we form an approximately sufficient statistic (based on wavelets) that can be represented using finite bits and does not result in rate-loss. The number of bits could still be more than $\ell$, and therefore, we use simulate-and-infer to generate samples from the statistic. Thus, loss of information due to communication constraints only happens in the last step, when we use multiple samples to simulate a sample from the sufficient statistic; the quantization part is just for efficient finite representation.

**Sobolev spaces and Fourier bases.** A first approach that we tried for Sobolev spaces was to use the Fourier basis, a natural choice for a Sobolev space. However, all our attempts led to a suboptimal performance either in the dependence on $\ell$ (we were not able to get the exponential $2^{-\ell}$ dependence) or the exponent of $n$ (*i.e.*, when we tried to get $2^{-\ell}$ dependence, this resulted in a suboptimal exponent of $n$). Somewhat surprisingly, the more general wavelet-based approach described above gives us tight bounds for Sobolev space as well, since Sobolev space $\mathcal{S}(\beta) = \mathcal{B}(2,2,\beta)$. Thus, it seems that it is necessary to use the wavelet representation even for Sobolev spaces to get an appropriately "small" statistic – sparsity of wavelets is very useful for inference under communication constraints.

**Organization.** After discussing prior works (Section 1.3) and preliminaries (Section 2), we describe our estimation algorithms in Section 3. In Section 3.1, we present the nonadaptive *single-level estimator* (comprising Algorithm 2 and Algorithm 3), which achieves guarantees of Theorem 1.2

(proof in Appendix B). In Section 3.2, we present the adaptive *multi-level estimator* (comprising Algorithm 4 and Algorithm 5), which achieves guarantees of Theorem 1.3 (proof in Appendix C). Finally, in Section 4, we briefly discuss our approach to proving lower bounds of Theorem 1.1, with details deferred to Appendix D. Appendices can be found in the Supplementary.

## 1.3 Prior work

The seminal work of Donoho, Johnstone, Kerkyacharian, and Picard [10] proposed wavelet estimators for Besov class in the centralized setting, and showed that these estimators achieve near-optimal rates of convergence (up to logarithmic factors),

$$\mathcal{L}_r^*(n, \infty, p, q, s) \asymp \begin{cases} n^{-\frac{rs}{2s+1}}, & r < (2s+1)p, \\ n^{-\frac{r(s-1/p+1/r)}{2(s-1/p)+1}}, & r \geq (2s+1)p. \end{cases} \tag{3}$$

Their results highlight the fact that linear estimators are inherently suboptimal for estimation with respect to $\mathcal{L}_r$ losses, when $r$ is large; that is, some nonlinearity in the estimator is required to achieve optimal rates for $r > p$. In particular, they show that nonlinearity in the form of *thresholding* achieves optimal rates for $r > p$ (see Section 2 below for details). Further, they use thresholding to design *adaptive* estimators that achieve near-optimal rates. These minmax rates exhibit the aforementioned *elbow effect*, where the error exponent is only piecewise linear, and changes slope at $r = (2s+1)p$. We refer the reader to [10] for a further discussion of these phenomena.

Butucea, Dubois, Kroll, and Saumard [7] recently extended these ideas to obtain near-rate optimal estimators for Besov spaces under local differential privacy constraints. Their adaptive estimator, as well as the information-theoretic lower bounds they establish, show that similar phenomena occur in the context of locally private nonparametric estimation. Our work, specifically the analysis of our adaptive estimator, draws upon some of the ideas of [7], with some crucial differences. In particular, the key ideas underlying our estimators – the wavelet-induced sparsity (Claim 3.1), the use of distributed simulation, and vector quantization – are neither present in nor applicable to the setting of [7] (where the introduction of random noise to ensure differential privacy effectively removes wavelet sparsity). Furthermore, our lower bounds even apply to interactive protocols, unlike the lower bounds from [7] which are restricted to the noninteractive setting.

In summary, our paper is the first to derive the counterpart of the nonparametric estimation results of [10, 7] under communication constraints, and shows that the analogue of the phenomena observed in [10] holds in the communication-constrained setting.

**Other work on distributed estimation.** We briefly discuss the related literature on distributed (communication-constrained) estimation problems. [18, 15, 16, 8] have studied the problem of distributed nonparametric function estimation (regression) under a Gaussian white noise model in a noninteractive setting with $n$ players, where each player observes an independent copy of the stochastic process $dY(t) = f(t)dt + (1/\sqrt{N})dW(t), 0 \leq t \leq 1$. Here $W(t)$ is the standard Wiener process, and $f$ is the function to be estimated. [18] derive minimax rates for $f$ in Sobolev space under $\mathcal{L}_2$ loss, where each player can send at most $\ell$ bits. [15] derive minimax rates for $f$ in the Besov space $\mathcal{B}(2, \infty, s)$ ("Sobolev type") under $\mathcal{L}_2$ loss, and $f$ in $\mathcal{B}(\infty, \infty, s)$ ("Hölder type") under $\mathcal{L}_\infty$ loss, where each player can send at most $\ell$ (assumed to be at least $\log N$) bits on average. Further, the paper proposes near-optimal adaptive estimators (based on Lepski's method) that adapt to the smoothness parameter $s$, provided that $s \in [s_{\min}, s_{\max})$, where $s_{\min}$ depends on $n, \ell$ and $s_{\max}$ can be arbitrary. [16] further study the problem of adaptivity for $\mathcal{B}(2, \infty, s)$ under $\mathcal{L}_2$ loss and $\mathcal{B}(\infty, \infty, s)$ under $\mathcal{L}_\infty$ loss, and answer the question of whether it is possible to design adaptive estimators (adapting to the smoothness parameter $s$) that attain centralized minimax rates while also having the expected communication budget nearly the same as that of a minimax optimal distributed estimator that knows $s$. The paper shows that this is possible for $\mathcal{B}(2, \infty, s)$ under $\mathcal{L}_2$ loss provided that $s$ is below a certain threshold, and is impossible for $\mathcal{B}(\infty, \infty, s)$ under $\mathcal{L}_\infty$ loss. [8] derive minimax rates for $f$ in Besov spaces $\mathcal{B}(p, q, s)$ (for $p \geq 2$) under $\mathcal{L}_2$ loss, where each player can send at most $\ell$ bits on average. In addition, they study the problem of adaptivity and characterize the minimax communication budget of adaptive estimators (adapting to parameters $p \geq 2, q > 1, s > 0$) that achieve centralized rates. The adaptive estimator proposed in this paper is based on thresholding, where the thresholding is done locally by each player.

Interestingly, when the communication budget $\ell$ is insufficient to achieve centralized rates, the minimax rates in these distributed nonparametric *function estimation* problems decay polynomially in

$\ell$, which is in contrast to the minimax rates we obtain for distributed nonparametric *density estimation* problem, where the decay is exponential in $\ell$.

For higher dimensions, [13] consider two-party estimation of Hölder smooth density functions in $d$ dimensions, where the two parties each observe a subset of coordinates and must communicate to estimate the density. They show that interactivity strictly helps over one-way communication.

We now discuss related works on distributed parametric estimation problems. [5] establish lower bounds on parametric density estimation, and on some restricted nonparametric families (Hölder classes) by bounding the Fisher information. [3] obtain upper and lower bounds for discrete distribution estimation; our algorithms leverage the concept of *distributed simulation* ("simulate-and-infer") introduced in that context. [12] derive lower bounds for various parametric estimation tasks, including discrete distributions and continuous (parametric) families such as high-dimensional Gaussians (including the sparse case). [2], building on [1] (which focused on learning and testing discrete distributions), developed a general technique to prove estimation lower bounds for parametric families; our lower bounds rely on their framework, by suitably extending it to handle the nonparametric case.

We note that there are other approaches for establishing lower bounds under communication constraints such as the early works [17], [14]; [11, 6], where bounds for specific inference problems under communication constraints were obtained; and [4], where Cramér–Rao bounds for this setting were developed. We found the general approach of [2] best fits our specific application, where we needed to handle interactive communication as well as a nonuniform prior on the parameter in the lower bound construction.

## 2 Preliminaries

Given two integers $m \leq n$, we write $[\![m, n]\!]$ for the set $\{m, m+1, \ldots, n\}$ and $[\![n]\!]$ for $[\![1, n]\!]$. For two sequences or functions $(a_n)_n, (b_n)_n$, we write $a_n \lesssim b_n$ if there exists a constant $C > 0$ (independent of $n$) such that $a_n \leq C b_n$ for all $n$, and $a_n \asymp b_n$ if both $a_n \lesssim b_n$ and $a_n \gtrsim b_n$. For a function $g$, $\mathrm{supp}(g)$ denotes the support of $g$.

Let $\mathcal{B}(p, q, s)$ be the Besov space with parameters $p, q, s$, where $1 \leq p, q \leq \infty$ and $s > 0$. We will write $\phi, \psi \in \mathcal{L}^2(\mathbb{R})$, respectively, for the father and mother wavelets generating the basis of the Besov space, and $\|f\|_{pqs}$ for the Besov norm of $f \in \mathcal{B}(p, q, s)$. We refer the reader to Appendix A in Supplementary for details on wavelets, Besov spaces and Besov norm.

**Assumptions on the density and wavelets.** We make the following assumptions on the density $f$:

1. $f$ is compactly supported: without loss of generality, $\mathrm{supp}(f) \subseteq [0, 1]$.
2. Besov norm of $f$ is bounded: without loss of generality, $\|f\|_{pqs} \leq 1$.

Our algorithm works with any father and mother wavelets $\phi$ and $\psi$ satisfying the following conditions:

1. $\phi$ and $\psi$ are $N$-regular, where $N > s$, and
2. $\mathrm{supp}(\phi), \mathrm{supp}(\psi) \subseteq [-A, A]$ for some integer $A > 0$ (which may depend on $N$).

As a concrete example, Daubechies' family of wavelets [9] satisfies these assumptions.

**Density estimation in centralized setting.** In the centralized setting, $X_1, \ldots, X_n$ from an unknown density $f \in \mathcal{B}(p, q, s)$ are accessible to the estimator. Let the wavelet expansion of $f$ be (see Appendix A.1 for details)

$$f = \sum_{k \in \mathbb{Z}} \alpha_{0,k} \phi_{0,k} + \sum_{j \geq 0} \sum_{k \in \mathbb{Z}} \beta_{j,k} \psi_{j,k} \tag{4}$$

where $\phi_{j,k}(x) = 2^{j/2} \phi(2^j x - k)$, $\psi_{j,k}(x) = 2^{j/2} \psi(2^j x - k)$. The wavelet basis satisfies the property that, for any $L, H \in \mathbb{Z}$ with $H \geq L$, we have $\sum_{k \in \mathbb{Z}} \alpha_{L,k} \phi_{L,k} + \sum_{j=L}^{H-1} \sum_{k \in \mathbb{Z}} \beta_{j,k} \psi_{j,k} = \sum_{k \in \mathbb{Z}} \alpha_{H,k} \phi_{H,k}$ (Fact A.1 in Supplementary). Note that for a given $j, k$, $\hat{\alpha}_{j,k} := \frac{1}{n} \sum_{i=1}^{n} \phi_{j,k}(X_i)$ is an unbiased estimate of $\alpha_{j,k}$. Thus, for some $H \in \mathbb{Z}_+$, an estimate of $f$ is

$$\hat{f}_{\mathrm{lin}} = \sum_{k \in \mathbb{Z}} \hat{\alpha}_{H,k} \phi_{H,k}, \quad \hat{\alpha}_{H,k} := \frac{1}{n} \sum_{i=1}^{n} \phi_{H,k}(X_i), \tag{5}$$

where $H$ is chosen depending on $n$ and parameters $p, q, s$ to minimize the (worst-case) $\mathcal{L}_r$ loss. This simple estimator (with appropriate choice of $H$) is rate-optimal when $1 \leq r \leq p$, but is sub-optimal when $r > p$ [10]. Moreover, setting $H$ requires knowing the Besov parameters $p, q, s$, which renders this estimator nonadaptive. The main contribution of [10] was to demonstrate that *thresholding* leads to estimators that are (i) near-optimal for every $r \geq 1$; (ii) adaptive, in the sense that the estimator does not use the values of parameters $p, q, s$ as long as $1/p < s < N$ for some $N \in \mathbb{N}$. For a given $L, H \in \mathbb{Z}_+, L \leq H$, a thresholded estimator outputs the estimate

$$\hat{f}_{\text{thresh}} = \sum_{k \in \mathbb{Z}} \hat{\alpha}_{L,k} \phi_{L,k} + \sum_{j=L}^{H} \sum_{k \in \mathbb{Z}} \tilde{\beta}_{j,k} \psi_{j,k}, \quad \tilde{\beta}_{j,k} = \hat{\beta}_{j,k} \mathbb{1}_{\{|\hat{\beta}_{j,k}| \geq t_j\}} \tag{6}$$

where $\hat{\alpha}_{L,k} := \frac{1}{n} \sum_{i=1}^{n} \phi_{L,k}(X_i)$, $\hat{\beta}_{j,k} = \frac{1}{n} \sum_{i=1}^{n} \psi_{j,k}(X_i)$, and $t_j$ is a fixed threshold proportional to $\sqrt{j/n}$; here, $L, H$ depend on $n$, but not on parameters $p, q, s$. Our proposed estimators draws upon these classical estimators.

## 3 Estimation Algorithms

We propose algorithms for density estimation under communication constraints that achieve optimal/near-optimal performance in terms of $n$ (number of players) and $\ell$ (number of bits each player can send). Designing a density estimator in the communication-constrained setting consists of: (i) specifying the sample-dependent $\ell$-bit message that a player sends to the referee; (ii) specifying the density estimate that the referee outputs based on the $\ell$-bit messages from the $n$ players. As in the unconstrained setting, we estimate $f$ by estimating its wavelet coefficients.

Our estimators consist of three ingredients: *wavelet-induced sparsity*, *vector quantization*, and *distributed simulation*.

**(i) Wavelet-induced sparsity.** Let the wavelet expansion of the density function $f$ be given by (4). For a given $J \in \mathbb{Z}_+$, partition the interval $[0, 1]$ into $2^J$ uniform bins as

$$[0,1] = \bigcup_{t=0}^{2^J-1} E_t^{(J)} \quad \text{where} \quad E_t^{(J)} := \begin{cases} [t2^{-J}, (t+1)2^{-J}) & \text{if } t \in [\![0, 2^J - 2]\!], \\ [1 - 2^{-J}, 1] & \text{if } t = 2^J - 1. \end{cases} \tag{7}$$

For a bin $E_t^{(J)}$, $t \in [\![0, 2^J - 1]\!]$, let

$$\mathcal{A}_t^{(J)} := \left\{ k \in \mathbb{Z} : E_t^{(J)} \cap \text{supp}(\phi_{J,k}) \text{ is non-empty} \right\}; \tag{8}$$

$$\mathcal{B}_t^{(J)} := \left\{ k \in \mathbb{Z} : E_t^{(J)} \cap \text{supp}(\psi_{J,k}) \text{ is non-empty} \right\}. \tag{9}$$

That is, for $x \in E_t^{(J)}$, we have $\phi_{J,k}(x) = 0$ for $k \notin \mathcal{A}_t^{(J)}$, and $\psi_{J,k}(x) = 0$ for $k \notin \mathcal{B}_t^{(J)}$. By "wavelet-induced sparsity," we mean the following:

**Claim 3.1.** *Let* $[0,1] = \bigcup_{t=0}^{2^J-1} E_t^{(J)}$ *as in* (7). *Then, for each* $t \in [\![0, 2^J - 1]\!]$,

$$|\mathcal{A}_t^{(J)}| \leq 2(A + 2), \quad |\mathcal{B}_t^{(J)}| \leq 2(A + 2),$$

*where* $A$ *is the assumed bound for points in the support of* $\phi$ *and* $\psi$.

The claim follows from the observation that $\phi_{J,k}$ (resp., $\psi_{J,k}$) is obtained by translating $\phi_{J,0}$ (resp., $\psi_{J,0}$) in steps of size $2^{-J}$, and that $\text{supp}(\phi_{J,k}) \subseteq [-A2^{-j}, A2^{-j}]$ (resp., $\text{supp}(\psi_{J,k}) \subseteq [-A2^{-j}, A2^{-j}]$).

**(ii) Vector quantization.** Consider the problem of designing a randomized algorithm that takes as input an arbitrary $x \in \mathbb{R}^d$ satisfying $\|x\|_\infty \leq B$, and outputs a random vector $Q(x) \in \mathbb{R}^d$ chosen from an alphabet of finite cardinality, such that $\mathbb{E}[Q(x)] = x$. Our vector quantization algorithm (Algorithm 1) achieves this, and is based on the following idea: Let $\mathcal{P}$ be a convex polytope with vertices $\{v_1, v_2, \ldots\}$ such that $\{x \in \mathbb{R}^d : \|x\|_\infty \leq B\} \subseteq \mathcal{P}$. Given $x$ (with $\|x\|_\infty \leq B$), express $x$ as convex combination of vertices of $\mathcal{P}$ (say, $x = \sum_i \theta_i v_i$) and output a random vertex $V$, where $V = v_i$ with probability $\theta_i$. Clearly, $\mathbb{E}[V] = x$.

Specifically, Algorithm 1 uses the polytope $\mathcal{P} = \mathcal{P}_{\mathcal{V}}$ formed by the vertex set $\mathcal{V} = \{\pm(Bd)e_1, \ldots, \pm(Bd)e_d\}$, where $e_i$ is the $i$-th standard basis vector (*i.e.*, $\mathcal{P}$ is the $\ell_1$ ball of radius $Bd$). Note that $|\mathcal{V}| = 2d$ and that $\{x \in \mathbb{R}^d : \|x\|_\infty \leq B\} \subseteq \mathcal{P}_{\mathcal{V}}$. This leads to the following claim.

**Claim 3.2.** *Given $x \in R^d$ with $\|x\|_\infty \leq B$ as input, Algorithm 1 outputs a random variable $Q(x) \in \mathcal{V}$ that is an unbiased estimate of $x$, with $|\mathcal{V}| = 2d$.*

**Remark.** A more direct approach to quantization would be to do it coordinate-wise, *i.e.*, quantize (independently) each coordinate to $\{-B, B\}$ with appropriate probability to make it unbiased. This can equivalently be seen as quantizing the vector using the $\ell_\infty$ ball (of radius $B$) as the polytope. Here, the alphabet size becomes $2^d$ instead of $2d$ in Algorithm 1; but, on the plus side, the coordinate-wise variance of the quantized vector becomes $\approx B^2$, instead of $\approx (Bd)^2$ in Algorithm 1. In our estimators, we will be quantizing vectors of constant length ($d$), so these dependencies on $d$ do not affect the rate (up to constants).

---

**Algorithm 1** Vector quantization

---

Let $\mathcal{V} = \{\pm(Bd)e_1, \ldots, \pm(Bd)e_d\}$. Label the vectors in $\mathcal{V}$ as $v_1, \ldots, v_{2d}$.
**Input:** $x \in \mathbb{R}^d$ with $\|x\|_\infty \leq B$.
 1: Write $x$ as convex combination of vectors in $\mathcal{V}$: $x = \sum_{i=1}^{2d} \theta_i v_i$.
 2: Choose $I \in \{1, \ldots, 2d\}$ randomly where $I = i$ with probability $\theta_i$ and **return** $Q(x) = v_I$.

---

**(iii) Distributed simulation.** The problem of *distributed simulation* is the following: There are $n$ players, each having an i.i.d. sample from an unknown $d$-ary distribution $\mathbf{p}$. Each player can only send $\ell$ bits to a central referee, where $\ell < \log d$. Can the referee simulate i.i.d. samples from $\mathbf{p}$ using $\ell$-bit messages from the players? [3] proposed a noninteractive communication protocol, using which the referee can simulate *one* sample from $\mathbf{p}$ using $\ell$-bit messages from $O(d/2^\ell)$ players. Moreover, the protocol is deterministic at the players, and only requires private randomness at the referee.

**Theorem 3.3** ([3]). *For any $\ell \geq 1$, the simulation protocol of [3], denoted $\mathrm{DISTRSIM}_\ell$, lets the referee simulate $\Omega(n2^\ell/d)$ i.i.d. samples from an unknown $d$-ary probability distribution $\mathbf{p}$ using $\ell$-bit messages from $n$ players, where each player holds an independent sample from $\mathbf{p}$.*

**Combining ideas.** We now discuss how the three ideas come together. To mimic the classical estimator (5), a player with sample $X$ would ideally like to communicate $\{\phi_{H,k}(X)\}_{k \in \mathbb{Z}}$, but cannot do so due to communication constraints. Wavelet-induced sparsity (Claim 3.1) ensures that communicating the bin (out of $2^H$ possible bins) in which $X$ lies is tantamount to identifying the set of at most $d := 2(A+2)$ indices $k$ for which $\phi_{H,k}(X)$ is possibly non-zero. Moreover, the player can quantize (unbiasedly) the vector containing values of $\phi_{H,k}(X)$ at these indices using Algorithm 1, whose output is one of $2d$ possibilities (Claim 3.2). Thus, overall, using an alphabet of size at most $2^H \times (2d) = O(2^H)$, a player can communicate an unbiased estimate of $\{\phi_{H,k}(X)\}_{k \in \mathbb{Z}}$. It can be shown that a density estimate based on these unbiased estimates from $n$ players still achieves centralized minmax rates (up to constants). However, if $2^\ell < 4(A+2)2^H$, the players cannot send these estimates directly to the referee. In this case, the players and the referee use the distributed simulation protocol $\mathrm{DISTRSIM}_\ell$ (Theorem 3.3), which, effectively, enables the referee to simulate $O(n2^\ell/2^H)$ i.i.d. realizations of unbiased estimates of $\{\phi_{H,k}(X)\}_{k \in \mathbb{Z}}$. The referee can now output a density estimate based on these simulated estimates. The degradation in minmax rates under communication constraints is due to the fact that the referee has only $O(n2^\ell/2^H)$ realizations of unbiased estimates of $\{\phi_{H,k}(X)\}_k$, instead of $n$.

We now give details of the idea outlined above. The resulting estimator (single-level estimator) is a communication-constrained version of the classical estimator given in (5). We then describe an adaptive estimator (multi-level estimator), which is a communication-constrained version of the classical adaptive estimator (6).

## 3.1 Single-level estimator

The $n$ players and the referee agree beforehand on the following: wavelet functions $\phi, \psi$; $H \in \mathbb{Z}_+$; partition $[0, 1] = \bigcup_{t=0}^{2^H - 1} E_t^{(H)}$ as in (7); collections of indices $\mathcal{A}_t^{(H)}$, $t \in [\![0, 2^H - 1]\!]$ as in (8). For every $t$, the indices in $\mathcal{A}_t^{(H)}$ are arranged in ascending order.

**Player's side.** Each player carries out two broad steps: (i) quantization; (ii) simulation.

---

**Algorithm 2** Single-level estimator (Players)

---

**Input:** Player-$i$ has input $X_i$, $i \in [\![n]\!]$.
 1: **for** $i = 1, \ldots, n$ **do**
 2:    Player-$i$ computes $Z_i := (B_i, Q(V_i))$, where: (i) $B_i$ is the bin in which $X_i$ lies; (ii) $Q(V_i)$ is
    an unbiased quantization of the vector $V_i := \left\{ 2^{-H/2} \phi_{H,k}(X_i) \right\}_{k \in \mathcal{A}_{B_i}^{(H)}}$.          ▷ **Quantization**
 3:    Player-$i$ computes $\ell$-bit message $Y_i$ corresponding to $Z_i$ as per DISTRSIM$_\ell$ (Theorem 3.3),
    and sends it to the referee.                               ▷ **Simulation**

---

The scaling by $2^{-H/2}$ in the definition of $V_i$ (line 2 in Algorithm 2) ensures that $\|V_i\|_\infty \le \|\phi\|_\infty$, which is a constant. This enables the use of Algorithm 1 to compute quantization of $V_i$. Overall, computing $Z_i = (B_i, Q(V_i))$ involves two quantizations: $B_i$ can be seen as a quantized version of $X_i \in [0,1]$; $Q(V_i)$ is a quantized version of $\{\phi_{H,k}(X_i)\}_k$. Moreover, for each $i \in [\![n]\!]$, $Z_i \in \mathcal{Z}^{(H)}$, where $\mathcal{Z}^{(H)} := [\![0, 2^H - 1]\!] \times \{\pm Be_1, \ldots, \pm Be_d\}$ (with $d \le 2(A+2)$, by Claim 3.1), so that $\left| \mathcal{Z}^{(H)} \right| \le 4(A+2)\, 2^H = O(2^H)$.

Thus, $Z_1, \ldots, Z_n$ are i.i.d. samples (since $X_1, \ldots, X_n$ are i.i.d.) from a $\left| \mathcal{Z}^{(H)} \right|$-ary distribution (call it $\mathbf{p}_{Z^{(H)}}$) distributed across $n$ players, where $\left| \mathcal{Z}^{(H)} \right| = O(2^H)$. Since a player can send only $\ell$ bits, player-$i$ cannot send $Z_i$ directly if $2^\ell < \left| \mathcal{Z}^{(H)} \right|$. In this case, player-$i$ computes an $\ell$-bit message $Y_i$ according to the distributed simulation protocol DISTRSIM$_\ell$, and sends $Y_i$ to the referee.

**Referee's side.** The referee, using the simulated i.i.d. samples from $\mathbf{p}_{Z^{(H)}}$, computes the density estimate similar to the classical estimate (5). This is possible because the $m = O(n2^\ell/2^H)$ simulated samples are, essentially, i.i.d. realizations of unbiased quantization of $\{\phi_{H,k}(X)\}_{k \in \mathbb{Z}}$.

---

**Algorithm 3** Single-level estimator (Referee)

---

**Input:** $Y_1, \ldots, Y_n$ ($\ell$-bit messages from $n$ players).
 1: From $Y_1, \ldots, Y_n$, referee obtains $m = O(n2^\ell/\left| \mathcal{Z}^{(H)} \right|) = O(n2^\ell/2^H)$ i.i.d. samples
    $Z_1', \ldots, Z_m' \sim \mathbf{p}_{Z^{(H)}}$ as per DISTRSIM$_\ell$, where $Z_i' = (B_i', Q_i') \in \mathcal{Z}^{(H)}$.
 2: **for** $i = 1, \ldots, m$ **do**
 3:    Referee computes

$$\widehat{\phi}_{H,k}^{(i)} := \begin{cases} 2^{H/2}\, Q_i'(k) & \text{if } k \in \mathcal{A}_{B_i'}^{(H)} \\ 0 & \text{otherwise,} \end{cases} \tag{10}$$

   where $Q_i'(k)$ is the entry in $Q_i'$ corresponding to index $k \in \mathcal{A}_{B_i'}^{(H)}$.     ▷ *Scaling by $2^{H/2}$ is to negate the scaling by $2^{-H/2}$ used in definition of $V_i$ on the players' side.*
 4: Referee outputs density estimate

$$\widehat{f} = \sum_{k \in \mathbb{Z}} \widehat{\alpha}_{H,k} \phi_{H,k}, \quad \text{where } \widehat{\alpha}_{H,k} = \frac{1}{m} \sum_{i=1}^m \widehat{\phi}_{H,k}^{(i)}, \; k \in \mathbb{Z}. \tag{11}$$

---

**Result.** For $H$ such that $2^H \asymp \min\{(n2^\ell)^{\frac{1}{2s+2}}, n^{\frac{1}{2s+1}}\}$, the single-level estimator recovers the guarantees in Theorem 1.2 (see Appendix B in Supplementary). The estimator is nonadaptive because setting $H$ requires knowing Besov parameter $s$. Further, note that the estimator is indeed noninteractive, as player-$i$'s message $Y_i$ does not depend on messages $Y_1, \ldots, Y_{i-1}$.

### 3.2 Multi-level estimator: An adaptive density estimator

The key observation in designing our multi-level estimator is that different coefficients need to be recovered with different accuracy. We enable this by dividing players into groups for estimating different coefficients, and using a different level of quantization for each group. This is in contrast to simply mimicking the classical adaptive estimator (6), which would suggest that a player with sample $X$ should quantize and communicate information about $\{\phi_{L,k}(X)\}_k$, $\left\{\{\psi_{J,k}(X)\}_k\right\}_{J \in [\![L,H]\!]}$. Instead, we do the following: Divide $n$ players into $H - L + 1$ groups of equal size (so, each

group has $n' = \frac{n}{H-L+1}$ players). Label the groups $L, L+1, \ldots, H$. Players in group-$L$ only focus on $\{\phi_{L,k}(X)\}_k$, $\{\psi_{L,k}(X)\}_k$. Players in group-$J$, $J \in [\![L+1, H]\!]$, only focus on $\{\psi_{J,k}(X)\}_k$. Moreover, players in group-$J$, $J \in [\![L, H]\!]$ quantize their sample $X$ using $2^J$ uniform bins. As before, by Claim 3.1, this is tantamount to identifying at most a constant number of indices for which the wavelet function evaluates to a non-zero value (since players in group-$J$ only consider $\phi_{J,k}$ or $\psi_{J,k}$). The player then quantizes the vector containing these values using Algorithm 1, before using distributed simulation.

The $n$ players and the referee agree beforehand on the following: wavelet functions $\phi, \psi$; $L, H \in \mathbb{Z}_+$; division of players into $H - L + 1$ groups. Further, for each $J \in [\![L, H]\!]$, the $n'$ players in group-$J$ and the referee agree on the following: partition $[0,1] = \bigcup_{t=0}^{2^J - 1} E_t^{(J)}$ as in (7); collection of indices $\mathcal{A}_t^{(J)}, \mathcal{B}_t^{(J)}$, $t \in [\![0, 2^J - 1]\!]$, as in (8), (9). For every $J, t$, the indices in $\mathcal{A}_t^{(J)}, \mathcal{B}_t^{(J)}$ are arranged in ascending order.

**Player's side.** Label players in group-$J$ as $(1, J), \ldots, (n', J)$. We denote by $X_{i,J}$ the sample with player-$(i, J)$. Essentially, players in group-$J$ run quantization and simulation steps as in the single-level algorithm (Algorithm 2), with $H$ replaced by $J$.

---

**Algorithm 4** Multi-level estimator (Players)

---

**Input:** Player-$(i, J)$ has input $X_{i,J}$, $i \in [\![n']\!]$, $J \in [\![L, H]\!]$ (where $n' = \frac{n}{H-L+1}$).

1: **for** $J = L, L+1, \ldots, H$ **do**
2:     **for** $i = 1, \ldots, n'$ **do**
3:         Player-$(i, J)$ computes $Z_{i,J} = (B_{i,J}, Q(V_{i,J}))$, where: (i) $B_{i,J}$ is the bin (out of $2^J$ bins) in which $X_{i,J}$ lies; (ii) $Q(V_{i,J})$ is an unbiased quantization of the vector $V_{i,J}$, where

$$V_{i,J} := \begin{cases} \left\{2^{-J/2}\psi_{J,k}(X_{i,J})\right\}_{k \in \mathcal{B}_{B_{i,J}}^{(J)}} & \text{if } J \in [\![L+1, H]\!], \\ \left\{2^{-L/2}\phi_{L,k}(X_{i,L})\right\}_{k \in \mathcal{A}_{B_{i,L}}^{(L)}} \oplus \left\{2^{-L/2}\psi_{L,k}(X_{i,L})\right\}_{k \in \mathcal{B}_{B_{i,L}}^{(L)}} & \text{if } J = L. \end{cases}$$

        ($\oplus$ denotes concatenation of two vectors.)           ▷ **Quantization**
4:         Player-$(i, J)$ computes $\ell$-bit message $Y_{i,J}$ corresponding to $Z_{i,J}$ as per $\text{DISTRSIM}_\ell$, and sends it to the referee.           ▷ **Simulation**

---

**Referee's side.** For players in group-$J$, $Z_{i,J} \in \mathcal{Z}^{(J)}$, where $\left|\mathcal{Z}^{(J)}\right| = O(2^J)$. Thus, after distributed simulation, referee obtains $m_J = O(n'2^\ell/2^J) = O(n2^\ell/(H-L+1)2^J)$ samples from players of group-$J$. Note that, higher the $J$, fewer the simulated samples; this dependence on $J$ of the number of samples available with the referee is one of the major differences between the classical and the distributed setting. Finally, using the simulated samples from players of every group, referee computes a density estimate similar to the adaptive classical estimator (6), with threshold value $t_J = \kappa\sqrt{J/m_J}$, for a constant $\kappa$.

**Result.** For $L, H$ satisfying $2^L \asymp \min\{(n2^\ell)^{\frac{1}{2(N+1)+2}}, n^{\frac{1}{2(N+1)+1}}\}$ and $2^H \asymp \min\{\frac{\sqrt{n2^\ell}}{\log(n2^\ell)}, \frac{n}{\log n}\}$, the multi-level estimator yields the guarantees in Theorem 1.3 (see Appendix C in Supplementary) as long as $s \in (1/p, N+1)$ (recall that $N$ is the regularity of the wavelet basis). Since $L, H$ do not depend on specific Besov parameters, the estimator is adaptive. Moreover, it is noninteractive.

## 4 Lower Bounds

We conclude with a description of our information-theoretic lower bounds (Theorem 1.1) for the minimax loss $\mathcal{L}_r^*(n, \ell, p, q, s)$, which applies to the broader class of interactive protocols (recall that our matching upper bounds are obtained by noninteractive ones); the details can be found in Appendix D in Supplementary. To derive lower bounds, we consider a family of probability distributions $\mathcal{P}$ parameterized by $\{-1, 1\}^d$ for some $d \in \mathbb{Z}_+$; that is, $\mathcal{P} = \{\mathbf{p}_z : z \in \{-1, 1\}^d\}$, where $\mathbf{p}_z$ has density $f_z$. Moreover, we specify a prior $\pi$ on $Z = (Z_1, \ldots, Z_d) \in \{-1, 1\}^d$, defined as $Z_i \sim \text{Rademacher}(\tau)$ independently for each $i \in [d]$, for some $\tau \in (0, 1/2]$. We then consider the following scenario:

---

**Algorithm 5** Multi-level algorithm (Referee)

---

**Input:** $\{Y_{i,J}\}_{i\in[\![n']\!], J\in[\![L,H]\!]}$ ($\ell$-bit messages from $n$ players).

1: **for** $J = L, L+1, \ldots, H$ **do**
2:      From $Y_{1,J}, \ldots, Y_{n',J}$, referee obtains $m_J = O(n2^\ell/(H-L+1)2^J)$ i.i.d. samples $Z'_{1,J}, \ldots, Z'_{m_J,J} \sim \mathbf{p}_{Z^{(J)}}$ as per $\text{DISTRSIM}_\ell$, where $Z'_{i,J} = (B'_{i,J}, Q'_{i,J}) \in \mathcal{Z}^{(J)}$.
3:      **for** $i = 1, \ldots, m_J$ **do**
4:          **if** $J = L$ **then**
5:              Referee computes $\left\{ \widehat{\phi}_{L,k}^{(i)} \right\}_{k\in\mathbb{Z}}$ as $\widehat{\phi}_{L,k}^{(i)} := \begin{cases} 2^{L/2}\, Q'_{i,L}(k) & \text{if } k \in \mathcal{A}_{B'_{i,L}}^{(L)} \\ 0 & \text{otherwise.} \end{cases}$
6:              Referee computes $\left\{ \widehat{\psi}_{J,k}^{(i)} \right\}_{k\in\mathbb{Z}}$ as $\widehat{\psi}_{J,k}^{(i)} := \begin{cases} 2^{J/2}\, Q'_{i,J}(k) & \text{if } k \in \mathcal{B}_{B'_{i,J}}^{(J)} \\ 0 & \text{otherwise.} \end{cases}$

7: Referee outputs density estimate

$$\widehat{f} = \sum_k \widehat{\alpha}_{L,k}\phi_{L,k} + \sum_{J=L}^H \sum_k \tilde{\beta}_{J,k}\psi_{J,k}, \tag{12}$$

where $\widehat{\alpha}_{L,k} = \frac{1}{m_L}\sum_{i=1}^{m_L}\widehat{\phi}_{L,k}^{(i)}$, $\widehat{\beta}_{J,k} = \frac{1}{m_J}\sum_{i=1}^{m_J}\widehat{\psi}_{J,k}^{(i)}$, $\tilde{\beta}_{J,k} = \widehat{\beta}_{J,k}\mathbb{1}_{\left\{|\widehat{\beta}_{J,k}|\geq t_J := \kappa\sqrt{J/m_J}\right\}}$.

---

For $Z \sim \pi$, let $X_1, \ldots, X_n$ be i.i.d. samples from $\mathbf{p}_Z$ distributed across $n$ players. Let $Y_1, \ldots, Y_n$ be $\ell$-bit messages sent by the players (possibly interactively) to the referee. Denote by $\mathbf{p}_{+i}^{Y^n}$ (resp. $\mathbf{p}_{-i}^{Y^n}$) the joint distribution of $Y_1, \ldots, Y_n$, given $Z_i = 1$ (resp. $Z_i = -1$). That is,

$$\mathbf{p}_{+i}^{Y^n} = \frac{1}{\tau}\sum_{z:z_i=1}\pi(z)\mathbf{p}_z^{Y^n}, \quad \mathbf{p}_{-i}^{Y^n} = \frac{1}{1-\tau}\sum_{z:z_i=-1}\pi(z)\mathbf{p}_z^{Y^n}, \tag{13}$$

where $\mathbf{p}_z^{Y^n}$ is the joint distribution of $Y_1, \ldots, Y_n$, given $Z = z$.

In this scenario, we analyze the "average discrepancy" $\frac{1}{d}\sum_{i=1}^d \mathrm{d}_{\mathrm{TV}}\left(\mathbf{p}_{-i}^{Y^n}, \mathbf{p}_{+i}^{Y^n}\right)$, where $\mathrm{d}_{\mathrm{TV}}(\mathbf{p},\mathbf{q})$ denotes the total variation distance between $\mathbf{p}$ and $\mathbf{q}$. On the one hand, a result from [2] gives us an *upper bound* on this average discrepancy as a function of $n$ and $\ell$ which holds for *any* interactive protocol generating $Y_1, \ldots, Y_n$ (Theorem D.3 in Supplementary). On the other hand, we derive a *lower bound* on average discrepancy (as a function of the error rate $\varepsilon$) as follows: Consider a communication-constrained density estimation algorithm (possibly interactive) which outputs $\hat{f}$ satisfying $\sup_{f\in\mathcal{B}(p,q,s)}\mathbb{E}_f\left[\|\hat{f}-f\|_r^r\right] \leq \varepsilon^r$. We show that one can use the messages $Y_1, \ldots, Y_n$ generated by this algorithm to solve, for each $i \in [\![d]\!]$, the binary hypothesis testing problem of deciding whether $Z_i = 1$ or $Z_i = -1$. This, in turn, implies a lower bound on $\frac{1}{d}\sum_{i=1}^d \mathrm{d}_{\mathrm{TV}}\left(\mathbf{p}_{-i}^{Y^n}, \mathbf{p}_{+i}^{Y^n}\right)$. Putting together the upper and lower bounds on average discrepancy gives us a lower bound on $\varepsilon$.

The parameterized family of distributions $\mathcal{P}$ we consider is constructed as follows: Let $f_0$ be a function supported on $[0,1]$. Let $I_1, \ldots, I_d \subseteq [0,1]$ be mutually disjoint intervals of equal length. Let $\psi_i$ be a "bump" function supported on interval $I_i$, where $\psi_i$'s are all translations of the same bump function. Then, for $z = (z_1, \ldots, z_d) \in \{-1,1\}^d$, we define $p_z$ to be a probability distribution with density $f_z$, defined as the "baseline" $f_0$ perturbed by adding (a rescaling of) the bump $\psi_i$ according to the value of $z_i$. In more detail, to get the desired lower bounds, we distinguish two cases depending on whether $r < (s+1)p$, and construct two families of distributions: $\mathcal{P}_1$ (when $r < (s+1)p$) and $\mathcal{P}_2$ (when $r \geq (s+1)p$). For $\mathcal{P}_1$, we use a uniform prior on $Z = (Z_1, \ldots, Z_d)$, *i.e.*, $Z$ has independent $\texttt{Rademacher}(1/2)$ coordinates, and set $f_z = f_0 + \gamma\sum_{i=1}^d z_i\psi_i$ for some suitably small parameter $\gamma > 0$. That is, the baseline density $f_0$ has disjoint bumps, which are either $\psi_i$ or $-\psi_i$ depending on the value of $z_i$. (See Appendix D.3 for details.) For $\mathcal{P}_2$, we use a non-uniform ("sparse") prior on $Z$, where $Z$ has independent $\texttt{Rademacher}(1/d)$ coordinates, and set $f_z = f_0 + \gamma\sum_{i=1}^d (1+z_i)\psi_i$ (so that bump $\psi_i$ only appears if $Z_i = 1$). (See Appendix D.4 for details.) Applying to these constructions the method described above allows us to derive the lower bounds of Theorem 1.1.

## Acknowledgments and Disclosure of Funding

Aditya Vikram Singh is supported by Prime Minister's Research Fellowship. Jayadev Acharya is supported by NSF-CCF- 1846300 (CAREER), NSF-CCF-1815893, and a Google Faculty Fellowship.

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
