# Optimal Rates for Nonparametric Density Estimation under Communication Constraints: Appendix

Jayadev Acharya
acharya@cornell.edu

Clément L. Canonne
clement.canonne@sydney.edu.au

Aditya Vikram Singh
adityavs@iisc.ac.in

Himanshu Tyagi
htyagi@iisc.ac.in

# Contents

**Organization.** In Appendix A, we define Besov spaces and state some useful facts about them. In Appendix B, we analyze the single-level estimator and prove Theorem 1.2. In Appendix C, we analyze the multi-level estimator and prove Theorem 1.3. Finally, in Appendix D, we give details of our lower bound technique and prove Theorem 1.1.

# A  Preliminaries

In this section, we formally define Besov spaces and provide some standard facts about them. We then state two useful probabilistic inequalities. Finally, we recall the assumptions that we make on the density function $f$.

## A.1  Besov spaces

Our exposition here mainly derives from [4, 5]. We start with a discussion on wavelets.

**Wavelets.** A wavelet basis for $\mathcal{L}^2(\mathbb{R})$ is generated using two functions: $\phi$ (father wavelet) and $\psi$ (mother wavelet). The main feature that distinguishes wavelet basis from the Fourier basis is that the functions $\phi$ and $\psi$ can have compact support. More precisely, there exists a function $\phi\colon \mathbb{R} \to \mathbb{R}$ such that

1. $\{\phi(\cdot - k) : k \in \mathbb{Z}\}$ forms an orthonormal family of $\mathcal{L}^2(\mathbb{R})$. Let $V_0 = \mathrm{span}\,\{\phi(\cdot - k) : k \in \mathbb{Z}\}$.

2. For $j \in \mathbb{Z}$, let $V_j = \mathrm{span}\,\{\phi_{j,k} : k \in \mathbb{Z}\}$, where $\phi_{j,k}(x) = 2^{j/2}\phi(2^j x - k)$. Then $V_j \subset V_{j+1}$.

3. $\phi \in \mathcal{L}^2(\mathbb{R})$, $\int \phi(x)dx = 1$.

   Remark: Conditions 1,2,3 ensure that $\cap_{j \in \mathbb{Z}} V_j = \{0\}$ and $\cup_{j \in \mathbb{Z}} V_j = \mathcal{L}^2(\mathbb{R})$.

4. $\phi$ satisfies the following regularity conditions for a given $N \in \mathbb{Z}_+$:

   (a) There exists a bounded non-increasing function $\Phi$ such that $\int \Phi(|x|)|x|^N dx < \infty$, and $|\phi(x)| \leq \Phi(|x|)$ almost everywhere.

   (b) $\phi$ is $N+1$ times (weakly) differentiable and $\phi^{(N+1)}$ satisfies $\mathrm{ess}\,\sup_x \sum_{k \in \mathbb{Z}} \left|\phi^{(N+1)}(x-k)\right| < \infty$.

   $\phi$ satisfying (a),(b) is said to be $N$-regular.

Let $W_j \subset \mathcal{L}^2(\mathbb{R})$ be a subspace such that $V_{j+1} = V_j \bigoplus W_j$ (i.e. $V_{j+1} = V_j + W_j$ and $V_j \cap W_j = \{0\}$). Then, there exists a function $\psi\colon \mathbb{R} \to \mathbb{R}$ such that

1. $\{\psi(\cdot - k) : k \in \mathbb{Z}\}$ forms an orthonormal basis of $W_0$.

2. $\mathrm{span}\,\{\psi_{j,k} : j \in \mathbb{Z}, k \in Z\} = \mathcal{L}^2(\mathbb{R})$, where $\psi_{j,k}(x) = 2^{j/2}\psi(2^j x - k)$.

3. $\psi$ satisfies the same regularity conditions as $\phi$.

For any $L \in \mathbb{Z}$, we can decompose $\mathcal{L}^2(\mathbb{R})$ as

$$\mathcal{L}^2(\mathbb{R}) = V_L \bigoplus W_L \bigoplus W_{L+1} \bigoplus \cdots.$$

That is, for any $f \in \mathcal{L}^2(\mathbb{R})$

$$f = \sum_{k \in \mathbb{Z}} \alpha_{L,k}\phi_{L,k} + \sum_{j \geq L}\sum_{k \in \mathbb{Z}} \beta_{j,k}\psi_{j,k}, \tag{1}$$

where

$$\alpha_{L,k} = \int f(x)\phi_{L,k}(x)dx, \quad \beta_{j,k} = \int f(x)\psi_{j,k}(x)dx$$

are called the wavelet coefficients of $f$, where the convergence is to be understood in the $\mathcal{L}^2$ sense in general; however, when $\phi, \psi$ satisfies the regularity condition, then the convergence holds in $\mathcal{L}^p$ sense for $p \in [1, \infty]$ (see Fact A.4). Moreover, for a father wavelet $\phi$, there is a canonical mother wavelet $\psi$ (Section 5.2 in [5]) corresponding to $\phi$.

**Besov spaces.** We now define Besov space $\mathcal{B}(p, q, s)$ with parameters $p, q, s$, where $1 \leq p, q \leq \infty$, $s > 0$. Let $\phi$ be a father wavelet satisfying properties (1)-(4) above, with $N > s$. Then,

$$f \in \mathcal{B}(p, q, s) \iff \|\alpha_{0\cdot}\|_p + \left( \sum_{j=0}^{\infty} \left( 2^{s + \frac{1}{2} - \frac{1}{p}} \|\beta_{j\cdot}\|_p \right)^q \right)^{1/q} < \infty \tag{2}$$

where $\|\alpha_{0\cdot}\|_p$ is the $\ell_p$ norm of the sequence $\{\alpha_{0,k}\}_{k\in\mathbb{Z}}$, $\|\beta_{j\cdot}\|_p$ is the $\ell_p$ norm of the sequence $\{\beta_{j,k}\}_{k\in\mathbb{Z}}$. The sequences $\{\alpha_{0,k}\}_{k\in\mathbb{Z}}, \{\beta_{j,k}\}_{k\in\mathbb{Z}}$ come from the wavelet expansion of $f$ using the father wavelet $\phi$ and the corresponding mother wavelet $\psi$. The definition (2) of $\mathcal{B}(p, q, s)$ is invariant to the choice of $\phi$ as long as $N > s$. For the purposes of defining Besov norm, we fix a particular $\phi, \psi$, where $\phi$ is $N$-regular with $N > s$. Then, the *Besov norm* of a function $f$ is defined as

$$\|f\|_{pqs} := \|\alpha_{0\cdot}\|_p + \left( \sum_{j=0}^{\infty} \left( 2^{s + \frac{1}{2} - \frac{1}{p}} \|\beta_{j\cdot}\|_p \right)^q \right)^{1/q}. \tag{3}$$

## A.2 Useful facts about Besov spaces

We record a few facts about Besov spaces that will be used in our analysis. Throughout, we assume that $f \in \mathcal{B}(p, q, s)$ with $\|f\|_{spq} \leq 1$ and $\mathrm{supp}(f) \subseteq [0, 1]$. The following fact is apparent from our discussion on wavelets.

**Fact A.1.** *Let the wavelet expansion of $f$ be as in (1). For $H \geq L$, define $f^{(H)} := \sum_{k\in\mathbb{Z}} \alpha_{L,k}\phi_{L,k} + \sum_{j=L}^{H-1} \sum_{k\in\mathbb{Z}} \beta_{j,k}\psi_{j,k}$. Then, $f^{(H)} = \sum_{k\in\mathbb{Z}} \alpha_{H,k}\phi_{H,k}$.*

Since $\mathrm{supp}(f) \subseteq [0, 1]$, and $\mathrm{supp}(\phi_{j,0}), \mathrm{supp}(\psi_{j,0}) \subseteq [-A2^{-j}, A2^{-j}]$, there is no overlap between $\mathrm{supp}(f)$ and $\mathrm{supp}(\phi_{j,k}), \mathrm{supp}(\psi_{j,k})$ for all but a finite number of indices $k$. In particular, we have the following.

**Fact A.2.** *Let the wavelet expansion of $f$ be as in (1). Then, for any given $j \in \mathbb{Z}_+$, there are $O(2^j)$ translation indices $k$ such that $\phi_{j,k}(x)$ or $\psi_{j,k}(x)$ is possibly non-zero, where $x \in \mathrm{supp}(f)$.*

For $f \in \mathcal{B}(p, q, s)$, it is clear from the definition of Besov norm that the wavelet coefficients must decay sufficiently fast. More precisely, we have the following.

**Fact A.3.** *If $f \in \mathcal{B}(p, q, s)$, then*

$$\lim_{j\to\infty} 2^{jp(s + \frac{1}{2} - \frac{1}{p})} \sum_{k\in\mathbb{Z}} |\beta_{j,k}|^p = 0$$

*and in particular there exists $C > 0$ such that*

$$\|\beta_j\|_p^p = \sum_{k\in\mathbb{Z}} |\beta_{j,k}|^p \leq C \cdot 2^{-jp(s + \frac{1}{2} - \frac{1}{p})}.$$

The next fact quantifies the approximation error when the wavelet expansion of $f$ is truncated.

**Fact A.4.** *Let $f^{(H)}$ be as in Fact A.1. Then, for $r \geq 1$,*

$$\left\|f^{(H)} - f\right\|_r \leq C \begin{cases} 2^{-Hs} & \text{if } r \leq p, \\ 2^{-H(s-1/p+1/r)} & \text{if } r > p. \end{cases}$$

The next fact (from equation (15) in [4]) gives a bound on $\|f\|_\infty$ when $\|f\|_{spq} \leq 1$.

**Fact A.5.** *Let $s > 1/p$. Then*

$$\|f\|_\infty \leq \left(1 - 2^{-(s-1/p)q'}\right)^{1/q'}$$

*where $1/q + 1/q' = 1$.*

Now, let $X_1, \ldots, X_n$ be independent samples from distribution with density $f$. For $j, k \in \mathbb{Z}$, define

$$\bar{\alpha}_{j,k} := \frac{1}{n}\sum_{i=1}^{n} \phi_{j,k}(X_i), \quad \bar{\beta}_{j,k} := \frac{1}{n}\sum_{i=1}^{n} \psi_{j,k}(X_i). \tag{4}$$

Observe that $\bar{\alpha}_{j,k}$ (resp., $\bar{\beta}_{j,k}$) is an unbiased estimate of $\alpha_{j,k}$ (resp., $\beta_{j,k}$). The following fact is from equation (16) in [4].

**Fact A.6.** *Let $n \geq 2^j$. Then, for $r \geq 1$,*

$$\mathbb{E}[|\bar{\alpha}_{j,k} - \alpha_{j,k}|^r] \leq Cn^{-r/2}, \quad \mathbb{E}\left[\left|\bar{\beta}_{j,k} - \beta_{j,k}\right|^r\right] \leq Cn^{-r/2}$$

*where $C$ is a constant that depends on $p, q, s, r, \phi, \psi$.*

We note another useful fact (obtained after setting $\beta = 0$ in equation (21) in [4]).

**Fact A.7.** *Let $g = \sum_{j=L}^{H} \sum_{k \in \mathbb{Z}} \widehat{g}_{j,k}\psi_{j,k}$, where $\widehat{g}_{j,k}$ is random. Then, for $r \geq 1$,*

$$\mathbb{E}[\|g\|_r^r] \leq C(H-L)^{(r/2-1)_+} \sum_{j=L}^{H} 2^{j(r/2-1)} \sum_{k \in \mathbb{Z}} \mathbb{E}[|\widehat{g}_{j,k}|^r]$$

*where $C$ is a constant that depends on $r$.*

## A.3 Useful probabilistic inequalities

**Theorem A.8** (Rosenthal's inequality [6])**.** *Let $X_1, \ldots, X_n$ be independent random variables such that $\mathbb{E}[X_i] = 0$. and $\mathbb{E}[|X_i|^r] < \infty$ for every $i$.*

1. *Suppose $\mathbb{E}\left[X_i^2\right] < \infty$ for every $i$. Then, for $1 \leq r \leq 2$,*

$$\mathbb{E}\left[\left|\sum_{i=1}^{n} X_i\right|^r\right] \leq \left(\sum_{i=1}^{n} \mathbb{E}\left[X_i^2\right]\right)^{r/2}.$$

   *(This just follows from concavity of $f(x) = x^{r/2}$ for $r \leq 2$.)*

*2. Suppose $\mathbb{E}[|X_i|^r] < \infty$ for every $i$. Then, for $r > 2$, there exists a constant $K_r$ depending only on $r$ such that*

$$\mathbb{E}\left[\left|\sum_{i=1}^{n} X_i\right|^r\right] \leq K_r \left\{\sum_{i=1}^{n} \mathbb{E}[|X_i|^r] + \left(\sum_{i=1}^{n} \mathbb{E}\left[X_i^2\right]\right)^{r/2}\right\}.$$

**Theorem A.9** (Bernstein's inequality)**.** *Let $X_1, \ldots, X_n$ be independent random variables such that $|X_i| \leq b$ almost surely, and $\mathbb{E}\left[X_i^2\right] \leq v_i$ for every $i$. Let $X := \sum_{i=1}^{n} X_i$ and $V := \sum_{i=1}^{n} v_i$. Then, for every $u \geq 0$,*

$$\Pr(|X - \mathbb{E}[X]| \geq u) \leq \exp\left(-\frac{u^2}{2(V + \frac{bu}{3})}\right)$$

## A.4  Assumptions

We recall here the assumptions we make on the density $f$:

1. $f$ is compactly supported: without loss of generality, $\mathrm{supp}(f) \subseteq [0, 1]$.

2. Besov norm of $f$ is bounded: without loss of generality, $\|f\|_{pqs} \leq 1$.

Our algorithm works with any father and mother wavelets $\phi$ and $\psi$ satisfying the following conditions:

1. $\phi$ and $\psi$ are $N$-regular, where $N > s$, and

2. $\mathrm{supp}(\phi), \mathrm{supp}(\psi) \subseteq [-A, A]$ for some integer $A > 0$ (which may depend on $N$).

As a concrete example, Daubechies' family of wavelets [3] satisfies these assumptions.

# B  Analysis of single-level estimator

Our goal is to upperbound the worst-case $\mathcal{L}^r$ loss $\mathbb{E}\left[\left\|\widehat{f} - f\right\|_r^r\right]$, where $\widehat{f}$ is the estimate output by the referee. Arguments and results in Sections B.1 and B.2 will be used in the analysis of the multi-level estimator as well.

## B.1  Coupling of simulated and ideal estimators

Denote by $\mathbf{p}$ the probability distribution corresponding to density $f$. Recall that $\mathbf{p}_{\mathcal{Z}(H)}$ is the distribution after quantization of samples from $\mathbf{p}$. Suppose the referee has $m$ samples from $\mathbf{p}_{\mathcal{Z}(H)}$ using which it outputs the estimate $\widehat{f}$. To compute $\mathbb{E}\left[\left\|\widehat{f} - f\right\|_r^r\right]$, consider the following statistically equivalent situation:

- There are $m$ i.i.d. samples $X_1, \ldots, X_m \sim \mathbf{p}$.

- For each $i \in [\![m]\!]$, let

$$\widehat{\phi}_{H,k}(X_i) = \begin{cases} 0 & \text{if } k \notin \mathcal{A}_{B_i}^{(H)}, \\ 2^{H/2} Q(V_i)(k) & \text{if } k \in \mathcal{A}_{B_i}^{(H)}, \end{cases} \tag{5}$$

  where $B_i$ is the bin $X_i$ lies in, $Q(V_i)$ is obtained by quantizing $\left\{2^{-H/2}\phi_{H,k}(X_i)\right\}_{k \in \mathcal{A}_{B_i}^{(H)}}$ using Algorithm 1, and $Q(V_i)(k)$ is the entry in $Q(V_i)$ corresponding to $k \in \mathcal{A}_{B_i}^{(H)}$. In other words, $\left\{\widehat{\phi}_{H,k}(X_i)\right\}_{k \in \mathbb{Z}}$ is the quantized version of $\{\phi_{H,k}(X_i)\}_{k \in \mathbb{Z}}$.

- Define $\widehat{f}$ as

$$\widehat{f} = \sum_{k \in \mathbb{Z}} \widehat{\alpha}_{H,k} \phi_{H,k}, \quad \text{where } \widehat{\alpha}_{H,k} = \frac{1}{m} \sum_{i=1}^{m} \widehat{\phi}_{H,k}(X_i), \ k \in \mathbb{Z}. \tag{6}$$

Then, computing the $\mathcal{L}^r$ loss for the single-level estimator is equivalent to computing the $\mathcal{L}^r$ loss for $\widehat{f}$ defined in (6). From here on, when we refer to $\widehat{f}$, we mean $\widehat{f}$ defined in (6). Now,

$$\mathbb{E}\left[\left\|\widehat{f} - f\right\|_r^r\right] \leq 2^{r-1}\left(\mathbb{E}\left[\left\|\widehat{f} - \bar{f}\right\|_r^r\right] + \mathbb{E}\left[\left\|\bar{f} - f\right\|_r^r\right]\right) \tag{7}$$

where $\bar{f}$ is defined as

$$\bar{f} = \sum_{k \in \mathbb{Z}} \bar{\alpha}_{H,k} \phi_{H,k} \quad \text{where} \quad \bar{\alpha}_{H,k} = \frac{1}{m} \sum_{i=1}^{m} \phi_{H,k}(X_i), \ k \in \mathbb{Z}. \tag{8}$$

Note that this is just the classical density estimate obtained using $m$ samples $X_1, \ldots, X_m$. The idea behind introducing this coupling is to facilitate analysis by bringing in the classical density estimate (8) and breaking up the $\mathcal{L}^r$ loss as in (7).

## B.2 Key lemmas

Since these lemmas will be used in the analysis of the multi-level estimator as well, we discuss them in more generality than would be required for only analyzing the single-level estimator. For $j, k \in \mathbb{Z}$, define

$$\widehat{\alpha}_{j,k} := \frac{1}{m} \sum_{i=1}^{m} \widehat{\phi}_{j,k}(X_i), \qquad \widehat{\beta}_{j,k} := \frac{1}{m} \sum_{i=1}^{m} \widehat{\psi}_{j,k}(X_i), \tag{9}$$

$$\bar{\alpha}_{j,k} := \frac{1}{m} \sum_{i=1}^{m} \phi_{j,k}(X_i), \qquad \bar{\beta}_{j,k} := \frac{1}{m} \sum_{i=1}^{m} \psi_{j,k}(X_i), \tag{10}$$

where $\left\{\widehat{\phi}_{j,k}(X_i)\right\}_{k \in \mathbb{Z}}$ and $\left\{\widehat{\psi}_{j,k}(X_i)\right\}_{k \in \mathbb{Z}}$ are quantized versions of $\{\phi_{j,k}(X_i)\}_{k \in \mathbb{Z}}$ and $\{\psi_{j,k}(X_i)\}_{k \in \mathbb{Z}}$, respectively (as in (5), with $H$ replaced by $j$; $\phi$ replaced by $\psi$, and $\mathcal{A}_{B_i}^{(j)}$ replaced by $\mathcal{B}_{B_i}^{(j)}$ in the case of $\widehat{\beta}_{j,k}$). The following claim bounds the error between quantized and unquantized (classical) estimates of wavelet coefficients.

**Claim B.1** (Error between quantized and unquantized estimates). *For $r \geq 1$, we have*

$$\mathbb{E}[|\widehat{\alpha}_{j,k} - \bar{\alpha}_{j,k}|^r] \leq C \begin{cases} \frac{1}{m^{r/2}}, & \text{if } r \in [1,2], \\ \frac{2^{j(r/2-1)}}{m^{r-1}} + \frac{1}{m^{r/2}}, & \text{if } r > 2, \end{cases}$$

*for a constant $C$. The same bound holds for $\mathbb{E}\left[\left|\widehat{\beta}_{j,k} - \bar{\beta}_{j,k}\right|^r\right]$ as well.*

*Proof.* For a given $j, k$,

$$\mathbb{E}[|\widehat{\alpha}_{j,k} - \bar{\alpha}_{j,k}|^r] = \mathbb{E}\left[\left|\frac{1}{m} \sum_{i=1}^{m} \left(\widehat{\phi}_{j,k}(X_i) - \phi_{j,k}(X_i)\right) \mathbb{1}_{\left\{\mathcal{A}_{B_i}^{(j)} \ni k\right\}}\right|^r\right] = \frac{1}{m^r} \mathbb{E}\left[\left|\sum_{i=1}^{m} Y_{ik}\right|^r\right]$$

where

$$Y_{ik} := \left(\widehat{\phi}_{j,k}(X_i) - \phi_{j,k}(X_i)\right) \mathbb{1}_{\left\{k \in \mathcal{A}_{B_i}^{(j)}\right\}}.$$

Note that, since the quantization is unbiased, we have $\mathbb{E}[Y_{ik}] = 0$. Moreover, $|Y_{ik}| \lesssim 2^{j/2}$ almost surely. We first consider the case $r > 2$. Then, by Rosenthal's inequality (Theorem A.8),

$$\mathbb{E}\left[\left|\sum_{i=1}^{m} Y_{ik}\right|^r\right] \lesssim (2^{j/2})^{(r-2)} \sum_{i=1}^{m} \mathbb{E}\left[Y_{ik}^2\right] + \left(\sum_{i=1}^{m} \mathbb{E}\left[Y_{ik}^2\right]\right)^{\frac{r}{2}} = 2^{j(\frac{r}{2}-1)} m \, \mathbb{E}\left[Y_{1k}^2\right] + m^{\frac{r}{2}} \mathbb{E}\left[Y_{1k}^2\right]^{\frac{r}{2}} \tag{11}$$

Moreover,

$$\mathbb{E}\left[Y_{ik}^2\right] = \mathbb{E}\left[\left(\widehat{\phi}_{j,k}(X_i) - \phi_{j,k}(X_i)\right)^2 \mathbb{1}_{\left\{k \in \mathcal{A}_{B_i}^{(j)}\right\}}\right] \lesssim 2^j \Pr\left(k \in \mathcal{A}_{B_i}^{(j)}\right).$$

Now, note that

$$\Pr\left(k \in \mathcal{A}_{B_i}^{(j)}\right) = \Pr(X_i \in \text{supp}(\phi_{j,k})) \leq \frac{2A}{2^j} \|f\|_\infty \lesssim \frac{1}{2^j} \quad \text{(using Fact A.5)}$$

which gives

$$\mathbb{E}\left[Y_{ik}^2\right] \lesssim 1.$$

Substituting this in (11), we get the desired result when $r > 2$. For $r \in [1,2]$, using part (1) of Theorem A.8, only the second term in (11) remains. This gives the result for $r \in [1,2]$. The proof for $\mathbb{E}\left[\left|\widehat{\beta}_{j,k} - \bar{\beta}_{j,k}\right|^r\right]$ is analogous. $\qquad \square$

The claim above, combined with Fact A.6, lets us bound the error between quantized estimates and true coefficients as follows.

**Claim B.2** (Error between quantized estimates and true coefficients). *Let $m \geq 2^j$. Then, for $r \geq 1$, we have*

$$\mathbb{E}[|\widehat{\alpha}_{j,k} - \alpha_{j,k}|^r] \leq \frac{C}{m^{r/2}}, \qquad \mathbb{E}\left[\left|\widehat{\beta}_{j,k} - \beta_{j,k}\right|^r\right] \leq \frac{C}{m^{r/2}}$$

*for a constant $C$.*

*Proof.* Note that

$$\mathbb{E}[|\widehat{\alpha}_{j,k} - \alpha_{j,k}|^r] \leq 2^{r-1}(\mathbb{E}[|\widehat{\alpha}_{j,k} - \bar{\alpha}_{j,k}|^r] + \mathbb{E}[|\bar{\alpha}_{j,k} - \alpha_{j,k}|^r]).$$

The first term can be handled with Claim B.1. The second term can be bound using Fact A.6. Overall, for $r > 2$, we get

$$\mathbb{E}[|\widehat{\alpha}_{j,k} - \alpha_{j,k}|^r] \lesssim \frac{2^{j(r/2-1)}}{m^{r-1}} + \frac{1}{m^{r/2}} \leq \frac{2}{m^{r/2}},$$

since $m \geq 2^j$. We get the same bound for $r \in [1, 2]$. The result follows. The bound on $\mathbb{E}\left[\left|\widehat{\beta}_{j,k} - \beta_{j,k}\right|^r\right]$ is obtained in the same way. $\qquad \square$

Since, for any $j$, there are $O(2^j)$ translations $k$ for which the coefficients are non-zero (Fact A.2), Claim B.2 readily implies the corollary below.

**Corollary B.3.** *Let $m \geq 2^j$. Then, for $r \geq 1$ and a constant $C$., we have*

$$\sum_{k \in \mathbb{Z}} \mathbb{E}[|\widehat{\alpha}_{j,k} - \alpha_{j,k}|^r] \leq C \frac{2^j}{m^{r/2}},$$

$$\sum_{k \in \mathbb{Z}} \mathbb{E}\left[\left|\widehat{\beta}_{j,k} - \beta_{j,k}\right|^r\right] \leq C \frac{2^j}{m^{r/2}}.$$

## B.3   Analyzing the error

For the single-level estimator

$$\mathbb{E}\left[\left\|\widehat{f} - f\right\|_r^r\right] = \mathbb{E}\left[\left\|\widehat{f} - f^{(H)} + f^{(H)} - f\right\|_r^r\right]$$

$$\leq 2^{r-1}\left(\mathbb{E}\left[\left\|\widehat{f} - f^{(H)}\right\|_r^r\right] + \left\|f - f^{(H)}\right\|_r^r\right), \qquad (12)$$

where $f^{(H)} = \sum_{k \in \mathbb{Z}} \alpha_{H,k} \phi_{H,k}$. Now, from Fact A.4, we have $\left\|f - f^{(H)}\right\|_r^r \lesssim 2^{-Hr\sigma}$, where

$$\sigma = \begin{cases} s & \text{if }, r \leq p, \\ (s - 1/p + 1/r), & \text{if } r > p. \end{cases}$$

Moreover,

$$\mathbb{E}\left[\left\|\widehat{f} - f^{(H)}\right\|_r^r\right] = \mathbb{E}\left[\left\|\sum_{k \in \mathbb{Z}} (\widehat{\alpha}_{H,k} - \alpha_{H,k}) \phi_{H,k}\right\|_r^r\right]$$

$$\lesssim 2^{H(r/2-1)} \sum_{k \in \mathbb{Z}} \mathbb{E}[|\hat{\alpha}_{H,k} - \alpha_{H,k}|^r] \qquad \text{(Fact A.7)}$$

$$\lesssim 2^{H(r/2-1)} \frac{2^H}{m^{r/2}} \qquad \text{(Corollary B.3)}$$

$$= \left(\frac{2^H}{m}\right)^{r/2}.$$

(In our case, $m, H$ will be such that $m \geq 2^H$ holds, which is why we can use Corollary B.3.) Substituting these in (12), we get

$$\mathbb{E}\left[\left\|\widehat{f} - f\right\|_r^r\right] \lesssim 2^{-Hr\sigma} + \left(\frac{2^H}{m}\right)^{r/2}.$$

Recall that $m$ is the number of quantized samples available with the referee, where, for a constant $C$,

$$m = \begin{cases} n & \text{if } 2^H \lesssim 2^\ell \text{ (no simulation required)} \\ Cn2^\ell/2^H & \text{if } 2^H \gtrsim 2^\ell \text{ (after simulation).} \end{cases}$$

In other words (ignoring constant $C$) $m = \frac{n2^\ell}{2^H \vee 2^\ell}$, where $a \vee b = \max\{a, b\}$. Thus,

$$\mathbb{E}\left[\left\|\widehat{f} - f\right\|_r^r\right] \lesssim 2^{-Hr\sigma} + \left(\frac{2^H(2^H \vee 2^\ell)}{n2^\ell}\right)^{r/2} \leq 2^{-Hr\sigma} + \left(\frac{2^{2H}}{n2^\ell}\right)^{r/2} + \left(\frac{2^H}{n}\right)^{r/2}.$$

Setting $H$ such that

$$2^H = (n2^\ell)^{\frac{1}{2s+2}} \wedge n^{\frac{1}{2s+1}}$$

gives us

$$\mathbb{E}\left\|\widehat{f} - f\right\|_r^r \lesssim (n2^\ell)^{-\frac{r\sigma}{2\sigma+2}} \vee n^{-\frac{r\sigma}{2\sigma+1}}.$$

For $r \leq p$, $\sigma = s$, which proves Theorem 1.2 in the main paper. $\qquad \square$

# C    Analysis of multi-level estimator

Our goal is to upper bound the worst-case $\mathcal{L}^r$ loss $\mathbb{E}\left[\left\|\widehat{f} - f\right\|_r^r\right]$, where $\widehat{f}$ is the estimate output by the referee. We proceed by describing the coupling in the multi-estimator setting.

## C.1    Coupling of simulated and ideal estimators

Denote by **p** the probability distribution corresponding to density $f$. Recall that we divide the players into $H - L + 1$ groups. Suppose the referee obtains $m_J$ (quantized) samples from players in group-$J$. To compute $\mathbb{E}\left[\left\|\widehat{f} - f\right\|_r^r\right]$, consider the following statistically equivalent situation:

- For each $J \in [\![L, H]\!]$, there are $m_J$ i.i.d. samples $X_{1,J}, \ldots, X_{m_J, J} \sim \mathbf{p}$.

- For each $i \in [\![m_L]\!]$, let

$$\widehat{\phi}_{L,k}(X_{i,L}) = \begin{cases} 0 & \text{if } k \notin \mathcal{A}^{(L)}_{B_{i,L}}, \\ 2^{L/2} Q(V_{i,L})(k) & \text{if } k \in \mathcal{A}^{(L)}_{B_{i,L}}, \end{cases} \tag{13}$$

  where $B_{i,L}$ is the bin (out of $2^L$ bins) $X_{i,L}$ lies in, $Q(V_{i,L})$ is obtained by quantizing $\left\{2^{-L/2}\phi_{L,k}(X_{i,L})\right\}_{k \in \mathcal{A}^{(L)}_{B_{i,L}}}$ using Algorithm 1, and $Q(V_{i,L})(k)$ is the entry in $Q(V_{i,L})$ corresponding to $k \in \mathcal{A}^{(L)}_{B_{i,L}}$. In other words, $\left\{\widehat{\phi}_{L,k}(X_{i,L})\right\}_{k \in \mathbb{Z}}$ is the quantized version of $\{\phi_{L,k}(X_{i,L})\}_{k \in \mathbb{Z}}$.

- Similarly, for each $J \in [\![L, H]\!]$, for each $i \in [\![m_J]\!]$, let

$$\widehat{\psi}_{J,k}(X_{i,J}) = \begin{cases} 0 & \text{if } k \notin \mathcal{B}^{(J)}_{B_{i,J}}, \\ 2^{J/2} Q(V_{i,J})(k) & \text{if } k \in \mathcal{B}^{(J)}_{B_{i,J}}. \end{cases} \tag{14}$$

  That is, for each $J \in [\![L, H]\!]$, $\left\{\widehat{\psi}_{J,k}(X_{i,J})\right\}_{k \in \mathbb{Z}}$ is the quantized version of $\{\psi_{J,k}(X_{i,J})\}_{k \in \mathbb{Z}}$.

- Given thresholds $\{t_J\}_{J \in [\![L,H]\!]}$, define $\widehat{f}$ as

$$\widehat{f} = \sum_k \widehat{\alpha}_{L,k} \phi_{L,k} + \sum_{J=L}^{H} \sum_k \tilde{\beta}_{J,k} \psi_{J,k}, \tag{15}$$

  where

$$\widehat{\alpha}_{L,k} = \frac{1}{m_L} \sum_{i=1}^{m_L} \widehat{\phi}_{L,k}(X_{i,L}),$$

$$\tilde{\beta}_{J,k} = \widehat{\beta}_{J,k} \mathbb{1}_{\left\{|\widehat{\beta}_{J,k}| \geq t_J\right\}} \text{ with } \widehat{\beta}_{J,k} = \frac{1}{m_J} \sum_{i=1}^{m_J} \widehat{\psi}_{J,k}(X_{i,J}). \tag{16}$$

Then, computing $\mathcal{L}^r$ loss for the multi-level estimator is equivalent to computing $\mathcal{L}^r$ loss for $\widehat{f}$ defined in (15).

## C.2 Setting parameters

Since we want our multi-level estimator to be adaptive, the parameters $L, H$, and $\{t_J\}_{J \in [\![L,H]\!]}$ should not depend explicitly on Besov parameters. We set $L, H$ as

$$2^L := C\left((n2^\ell)^{\frac{1}{2(N+1)+2}} \wedge n^{\frac{1}{2(N+1)+1}}\right),$$

$$2^H := C'\left(\frac{\sqrt{n2^\ell}}{\log n2^\ell} \wedge \frac{n}{\log^2 n}\right)$$

where $C, C' > 0$ are two constants, sufficiently large and small, respectively. Also, note that, since players in group-$J$ have alphabet size $O(2^J)$, we have

$$m_J = \frac{n}{(H - L + 1)} \cdot \left(\frac{2^\ell}{2^J} \wedge 1\right) \asymp \frac{n2^\ell}{H(2^J \vee 2^\ell)}.$$

Note that the setting of $H$ implies both $H2^H \ll \sqrt{n2^\ell}$ and $H^2 2^H \ll n$, and consequently $m_J \geq J2^J$.

**Threshold values.** How should we set the threshold values $\{t_J\}_{J \in [\![L,H]\!]}$? Since we will pay a cost for the coefficients we zero out (increase in bias), we would like to choose $t_J$ as small as possible. But, in order to have reasonable concentration, we also need $t_J$ to satisfy, for every (sufficiently large) $\gamma > 0$,

$$\Pr\left(\left|\widehat{\beta}_{J,k} - \beta_{J,k}\right| \geq \gamma t_J\right) \lesssim 2^{-\gamma J}.$$

so that our estimates concentrate well around their true value, and we only zero them out wrongly with very small probability. Now, a natural approach to choose $t_J$ according to the constraint above would be to use Hoeffding's inequality, as $\widehat{\beta}_{J,k}$ is the empirical mean of $m_J$ unbiased estimates of $\beta_{J,k}$, each with magnitude $\lesssim 2^{J/2}$. One can check that this would lead to the setting of $t_J \asymp \sqrt{J2^J/m_J}$, which, unfortunately, is too big (by a factor of $2^{J/2}$) to give optimal rates. However, recall that the $m_J$ unbiased estimates, $\widehat{\psi}_{J,k}(X_{i,J})$, are not only such that $|\widehat{\psi}_{J,k}(X_{i,J})| \lesssim 2^{J/2}$; in many cases, they are actually zero, since $|\widehat{\psi}_{J,k}(X_{i,J})| \simeq 2^{J/2} \mathbb{1}_{\left\{k \in \mathcal{B}_{B_{i,J}}^{(J)}\right\}}$. This allows us to derive the following, improving upon the naïve use of Hoeffding's inequality.

**Lemma C.1.** *For $J \in [\![L, H]\!]$, setting $t_J := \sqrt{J/m_J}$, we have*

$$\Pr\left(\left|\widehat{\beta}_{J,k} - \beta_{J,k}\right| \geq \gamma t_J\right) \leq 2^{-\gamma J}$$

*for every $\gamma \geq 6A\|f\|_\infty$.*

*Proof.* Fix $J, k$, and consider any $i \in [\![m_J]\!]$. Since $\left|\widehat{\psi}_{J,k}(X_{i,J})\right| \leq b := 2^{J/2}$ and

$$\mathbb{E}\left[\widehat{\psi}_{J,k}(X_{i,J})^2\right] = 2^J \Pr\left(k \in \mathcal{B}_{B_{i,J}}^{(J)}\right) \leq 2^J \cdot \|f\|_\infty \cdot \frac{2A}{2^J} = 2A\|f\|_\infty := v$$

where the inequality follows from our assumption that $\text{supp}(\psi) \subseteq [-A, A]$. In particular, we have $v \asymp 1$. Recalling the definition of $\widehat{\beta}_{J,k}$ from (16), we can apply Bernstein's inequality (Theorem A.9) to obtain, for $t \geq 0$ and $\gamma \geq 3v$,

$$\Pr\left(\left|\widehat{\beta}_{J,k} - \beta_{J,k}\right| \geq \gamma t\right) \leq e^{-\frac{3\gamma^2 m_J t^2}{6v + 2b\gamma t}} = e^{-\frac{3}{2} \cdot \frac{\gamma m_J t^2}{\frac{3v}{\gamma} + 2^{J/2}t}} \leq e^{-\frac{3}{2} \cdot \frac{\gamma m_J t^2}{1 + 2^{J/2}t}} \leq 2^{-\frac{2\gamma m_J t^2}{1 + 2^{J/2}t}}.$$

Setting $t_J := \sqrt{\frac{J}{m_J}} \vee \frac{J2^{J/2}}{m_J}$, we get $\Pr\left(\left|\widehat{\beta}_{J,k} - \beta_{J,k}\right|\right) \leq 2^{-\gamma J}$. Finally, our setting of $H$ and $m_J$ together imply that $t_J := \sqrt{\frac{J}{m_J}}$, as (from our choice of parameters) $m_J \geq J2^J$ for all $J \leq H$. $\quad\square$

**Conclusion.** For constants $C, C', \kappa > 0$, the values of parameters are summarized below.

$$2^L := C\left((n2^\ell)^{\frac{1}{2(N+1)+2}} \wedge n^{\frac{1}{2(N+1)+1}}\right) \tag{17}$$

$$2^H := C'\left(\frac{\sqrt{n2^\ell}}{\log n2^\ell} \wedge \frac{n}{\log^2 n}\right) \tag{18}$$

$$m_J := \frac{n}{(H-L+1)} \cdot \left(\frac{2^\ell}{2^J} \wedge 1\right) \asymp \frac{n2^\ell}{H(2^J \vee 2^\ell)} \tag{19}$$

$$t_J := \kappa\sqrt{\frac{J}{m_J}} \tag{20}$$

As previously mentioned choices imply both $H2^H \ll \sqrt{n2^\ell}$ and $H^2 2^H \ll n$, and consequently $m_J \geq J2^J$, $J \in [\![L, H]\!]$.

## C.3   Analysing the error

Following the outline of Theorem 3 of [4] and Theorem 5.1 of [2], we will bound $\mathcal{L}_r$ loss as

$$\mathbb{E}\left[\left\|f - \widehat{f}\right\|_r^r\right] \leq 3^{r-1}(\text{bias}(f) + \text{linear}(f) + \text{details}(f)) \tag{21}$$

where

$$\text{bias}(f) = \mathbb{E}\left[\left\|f - \sum_{k\in\mathbb{Z}} \alpha_{H,k}\phi_{H,k}\right\|_r^r\right]$$

$$\text{linear}(f) = \mathbb{E}\left[\left\|\sum_{k\in\mathbb{Z}} (\widehat{\alpha}_{L,k} - \alpha_{L,k})\phi_{L,k}\right\|_r^r\right]$$

$$\text{details}(f) = \mathbb{E}\left[\left\|\sum_{J=L}^{H} \sum_{k\in\mathbb{Z}} (\widetilde{\beta}_{J,k} - \beta_{J,k})\psi_{J,k}\right\|_r^r\right]$$

and handle each of the three terms separately. Note that only the third term relates to thresholding.

### C.3.1   Linear and bias terms

**Linear term.** To bound $\text{linear}(f)$, we invoke Fact A.7 and Corollary B.3 as in the analysis of single-level estimator. This gives

$$\mathbb{E}\left[\left\|\sum_{k\in\mathbb{Z}} (\widehat{\alpha}_{L,k} - \alpha_{L,k})\phi_{L,k}\right\|_r^r\right] \lesssim 2^{L(\frac{r}{2}-1)} \sum_{k\in\mathbb{Z}} \mathbb{E}[|\widehat{\alpha}_{L,k} - \alpha_{L,k}|^r] \lesssim 2^{L(\frac{r}{2}-1)} \cdot \frac{2^L}{m_L^{r/2}} \tag{22}$$

$$= H^{\frac{r}{2}}\left(\left(\frac{2^{2L}}{n2^\ell}\right)^{r/2} \vee \left(\frac{2^L}{n}\right)^{r/2}\right)$$

$$\lesssim H^{\frac{r}{2}}\left((n2^\ell)^{-\frac{r(N+1)}{2(N+1)+2}} \vee n^{-\frac{r(N+1)}{2(N+1)+1}}\right) \leq H^{\frac{r}{2}}\left((n2^\ell)^{-\frac{rs}{2s+2}} \vee n^{-\frac{rs}{2s+1}}\right)$$

$$(23)$$

where the second-to-last inequality relies on our choice of $L$.

**Bias term.** To bound bias($f$), we use Fact A.4 to get, with $s' = s - 1/p + 1/r$,

$$\text{bias}(f) \leq C \cdot 2^{-Hs'r} \leq C' \cdot \left(\sqrt{\frac{\log^2(n2^\ell)}{n2^\ell}} \vee \frac{\log^2 n}{n}\right)^{r(s-1/p+1/r)}.$$

$$(24)$$

### C.3.2   Details term

To bound the term details($f$), we define, for $J \in [\![L, H]\!]$, the three sets of indices:

$$\widehat{\mathcal{I}}_J := \{k \in \mathbb{Z} : |\widehat{\beta}_{J,k}| > \kappa t_J\} \qquad \text{(estimate big: not thresholded)}$$
$$\mathcal{I}_J^s := \{k \in \mathbb{Z} : |\beta_{J,k}| \leq \tfrac{1}{2}\kappa t_J\} \qquad \text{(small coefficients)}$$
$$\mathcal{I}_J^b := \{k \in \mathbb{Z} : |\beta_{J,k}| > 2\kappa t_J\} \qquad \text{(big coefficients)}$$

We will partition the error according to these sets of indices, and argue about them separately. Specifically, we write

$$\text{details}(f) = \mathbb{E}\left[\left\|\sum_{J=L}^{H}\sum_{k \in \widehat{\mathcal{I}}_J \cap \mathcal{I}_J^s}(\widetilde{\beta}_{J,k} - \beta_{J,k})\psi_{J,k}\right\|_r^r\right] + \mathbb{E}\left[\left\|\sum_{J=L}^{H}\sum_{k \in \widehat{\mathcal{I}}_J \setminus \mathcal{I}_J^s}(\widetilde{\beta}_{J,k} - \beta_{J,k})\psi_{J,k}\right\|_r^r\right]$$

$$+ \mathbb{E}\left[\left\|\sum_{J=L}^{H}\sum_{k \in \mathcal{I}_J^b \setminus \widehat{\mathcal{I}}_J}\beta_{J,k}\psi_{J,k}\right\|_r^r\right] + \mathbb{E}\left[\left\|\sum_{J=L}^{H}\sum_{k \notin \mathcal{I}_J^b \cup \widehat{\mathcal{I}}_J}\beta_{J,k}\psi_{J,k}\right\|_r^r\right]$$

$$= E_{bs} + E_{bb} + E_{sb} + E_{ss}, we$$

the four errors coming from the "big-small," "big-big," "small-big," and"small-small" indices, respectively. Our analysis is along the lines of that in [2].

**The term $E_{bs}$.**   We can write

$$E_{bs} \lesssim H^{r/2}\sum_{J=L}^{H}2^{J(\frac{r}{2}-1)}\sum_{k \in \mathbb{Z}}\mathbb{E}\left[|\widehat{\beta}_{J,k} - \beta_{J,k}|^r \mathbb{1}_{k \in \widehat{\mathcal{I}}_J \cap \mathcal{I}_J^s}\right] \qquad \text{(Fact A.7)}$$

$$\lesssim H^{r/2}\sum_{J=L}^{H}2^{J(\frac{r}{2}-1)}\sum_{k \in \mathbb{Z}}\mathbb{E}\left[|\widehat{\beta}_{J,k} - \beta_{J,k}|^{2r}\right]^{\frac{1}{2}}\Pr\left(k \in \widehat{\mathcal{I}}_J \cap \mathcal{I}_J^s\right)^{\frac{1}{2}} \qquad \text{(Cauchy–Schwarz)}$$

$$\lesssim H^{r/2}\sum_{J=L}^{H}2^{J(\frac{r}{2}-1)}\sum_{k \in \mathbb{Z}}\mathbb{E}\left[|\widehat{\beta}_{J,k} - \beta_{J,k}|^{2r}\right]^{\frac{1}{2}}\Pr\left(|\widehat{\beta}_{J,k} - \beta_{J,k}| > \tfrac{\kappa}{2}t_J\right)$$

$$\lesssim H^{r/2}\sum_{J=L}^{H}2^{J(\frac{r}{2}-1)}\sum_{k \in \mathbb{Z}}\mathbb{E}\left[|\widehat{\beta}_{J,k} - \beta_{J,k}|^{2r}\right]^{\frac{1}{2}}2^{-\frac{\kappa}{2}J} \qquad \text{(Lemma C.1)}$$

$$\lesssim H^{r/2}\sum_{J=L}^{H}2^{J(\frac{r-\kappa}{2}-1)}\frac{2^J}{m_J^{r/2}}$$

where the last inequality follows from Claim B.2 and the $O(2^J)$-sparsity of coefficients (Fact A.2). Going forward, recalling our setting of $m_J$ we get

$$
\begin{aligned}
E_{bs} &\lesssim H^{r/2} \sum_{J=L}^{H} \frac{2^{J\frac{r-\kappa}{2}}}{m_J^{r/2}} \lesssim H^{r/2} \left(\frac{H}{n2^\ell}\right)^{r/2} \sum_{J=L}^{H} 2^{J\frac{r-\kappa}{2}} (2^J \vee 2^\ell)^{r/2} \\
&\lesssim H^{r/2} \left(\frac{H}{n2^\ell}\right)^{r/2} \sum_{J=L}^{H} 2^{J(r-\frac{\kappa}{2})} + H^{r/2} \left(\frac{H}{n}\right)^{r/2} \sum_{J=L}^{H} 2^{J\frac{r-\kappa}{2}} \\
&\lesssim H^{r/2} \left(\frac{H}{n2^\ell}\right)^{r/2} 2^{L(r-\kappa/2)} + H^{r/2} \left(\frac{H}{n}\right)^{r/2} 2^{L\frac{r-\kappa}{2}} \lesssim H^r n^{-r/2}
\end{aligned}
\tag{25}
$$

where the inequalities hold for $\kappa > 2r$.

**The term $E_{bb}$.** Turning to the term $E_{bb}$, we have

$$
\begin{aligned}
E_{bb} &= \Big\| \sum_{J=L}^{H} \sum_{k\in\mathbb{Z}} (\widehat{\beta}_{J,k} - \beta_{J,k}) \psi_{J,k} \mathbb{1}_{k\in\widehat{\mathcal{I}}_J \setminus \mathcal{I}_J^s} \Big\|_r^r \\
&\lesssim H^{r/2} \sum_{J=L}^{H} 2^{J(r/2-1)} \sum_{k\in\mathbb{Z}} \mathbb{E}\left[ |\widehat{\beta}_{J,k} - \beta_{J,k}|^r \mathbb{1}_{k\in\widehat{\mathcal{I}}_J \setminus \mathcal{I}_J^s} \right] \qquad \text{(Fact A.7)} \\
&\lesssim H^{r/2} \sum_{J=L}^{H} \frac{2^{J(r/2-1)}}{m_J^{r/2}} \sum_{k\in\mathbb{Z}} \mathbb{1}_{k\notin\mathcal{I}_J^s}
\end{aligned}
$$

using that $\mathbb{1}_{k\in\widehat{\mathcal{I}}_J \setminus \mathcal{I}_J^s} \leq \mathbb{1}_{k\notin\mathcal{I}_J^s}$ and Claim B.2. Using the definition of $\mathcal{I}_J^s$, for any nonnegative sequence $(\alpha_J)_J$ we can further bound this as

$$
\begin{aligned}
E_{bb} &\lesssim H^{r/2} \sum_{J=L}^{H} \frac{2^{J(r/2-1)}}{m_J^{r/2}} \sum_{k\in\mathbb{Z}} \mathbb{1}_{k\notin\mathcal{I}_J^s} |\beta_{J,k}|^{\alpha_J} (\kappa t_J/2)^{-\alpha_J} \\
&\leq H^{r/2} \sum_{J=L}^{H} \frac{2^{J(r/2-1)}}{m_J^{r/2}} (\kappa t_J/2)^{-\alpha_J} \sum_{k\in\mathbb{Z}} |\beta_{J,k}|^{\alpha_J} \\
&= H^{r/2} \sum_{J=L}^{H} 2^{\alpha_J} \kappa^{-\alpha_J} J^{-\frac{\alpha_J}{2}} \frac{2^{J(r/2-1)}}{m_J^{(r-\alpha_J)/2}} \sum_{k\in\mathbb{Z}} |\beta_{J,k}|^{\alpha_J} \\
&\leq H^{r/2} \sum_{J=L}^{H} \frac{2^{J(r/2-1)}}{m_J^{(r-\alpha_J)/2}} \sum_{k\in\mathbb{Z}} |\beta_{J,k}|^{\alpha_J}
\end{aligned}
\tag{26}
$$

the last inequality using $\kappa/2 \geq 1$ and $J \geq 1$ to simplify the expression a little. For now, we ignore the factor $H^{r/2}$ (we will bring it back at the end), and look at two cases:

- If $p > \frac{r}{s+1}$, we continue by writing

$$
E_{bb} \lesssim \sum_{J=L}^{H} \frac{2^{J(r/2-1)}}{m_J^{(r-\alpha_J)/2}} 2^{-J\alpha_J(s+\frac{1}{2}-\frac{1}{\alpha_J})} \asymp \sum_{J=L}^{H} m_J^{-\frac{r-\alpha_J}{2}} 2^{\frac{1}{2}J(r-(2s+1)\alpha_J)} = \sum_{J=L}^{H} 2^{J\frac{r}{2}} m_J^{-\frac{r}{2}} \cdot 2^{\frac{\alpha_J}{2}(\log m_J - J(2s+1))}
$$

where the first inequality, which holds for any $\alpha_J \in [0,p]$, uses the following bound on $\sum_k |\beta_{J,k}|^p$: For any $\alpha \in [0,p]$, we have from Fact A.3 and Hölder's inequality along with

the sparsity of coefficients (Fact A.2), that

$$\sum_{k\in\mathbb{Z}}|\beta_{J,k}|^{\alpha} = \sum_{k\in\mathbb{Z}}|\beta_{J,k}|^{\alpha}\mathbb{1}_{\{\beta_{J,k}\neq 0\}} \leq \left(\sum_{k\in\mathbb{Z}}|\beta_{J,k}|^{p}\right)^{\frac{\alpha}{p}}|\mathcal{B}_J|^{1-\frac{\alpha}{p}} \leq C^{\frac{\alpha}{p}}(2A+1)^{1-\frac{\alpha}{p}}\cdot 2^{-J\alpha(s+\frac{1}{2}-\frac{1}{p})}\cdot 2^{J(1-\frac{\alpha}{p})}$$

so that $\sum_{k\in\mathbb{Z}}|\beta_{J,k}|^{\alpha} \lesssim 2^{-J\alpha(s+\frac{1}{2}-\frac{1}{\alpha})}$, as in [2, Section C.2.3]. To bound the resulting sum, we need to choose $\alpha_J \in [0,p]$ for all $J$ in order to minimize the result. Since $m_J \propto 2^{J-\ell}\wedge 1$, the quantity

$$\log m_J - J(2s+1)$$

is decreasing in $J$, and thus becomes negative at some value $M$ (for simplicity, assumed to be an integer), such that[1]

$$2^M \asymp \left(\frac{n2^\ell}{H}\right)^{\frac{1}{2s+2}} \wedge \left(\frac{n}{H}\right)^{\frac{1}{2s+1}} \tag{27}$$

we see that we should set $\alpha_J := 0$ for $J \leq M$, and for $J > M$ set all $\alpha_J$ to some value $\alpha = \alpha(r,s)$ which will balance the remaining terms. With this choice, we can write

$$\sum_{J=L}^{H} m_J^{-\frac{r-\alpha_J}{2}}2^{\frac{1}{2}J(r-(2s+1)\alpha_J)} \leq \sum_{J=L}^{M} m_J^{-\frac{r}{2}}2^{J\frac{r}{2}} + \sum_{J=M}^{H} m_J^{-\frac{r-\alpha}{2}}2^{\frac{1}{2}J(r-(2s+1)\alpha)}$$

$$\leq \left(\frac{H}{n2^\ell}\right)^{\frac{r}{2}}\sum_{J=1}^{M}2^{Jr} + \left(\frac{H}{n}\right)^{\frac{r}{2}}\sum_{J=1}^{M}2^{J\frac{r}{2}}$$

$$+ \left(\frac{H}{n2^\ell}\right)^{\frac{r-\alpha}{2}}\sum_{J=M}^{H}2^{J(r-(s+1)\alpha)} + \left(\frac{H}{n}\right)^{\frac{r-\alpha}{2}}\sum_{J=M}^{H}2^{\frac{1}{2}J(r-(2s+1)\alpha)}$$

recalling for the second inequality that $m_J^{-1} \asymp \frac{H}{n}(\frac{2^J}{2^\ell}\vee 1)$. We can bound the first and second terms as

$$\left(\frac{H}{n2^\ell}\right)^{\frac{r}{2}}\sum_{J=1}^{M}2^{Jr} \leq \frac{2^r}{2^r-1}\left(\frac{H}{n2^\ell}\right)^{\frac{r}{2}}2^{rM}, \qquad \left(\frac{H}{n}\right)^{\frac{r}{2}}\sum_{J=1}^{M}2^{J\frac{r}{2}} \leq \frac{2^{r/2}}{2^{r/2}-1}\left(\frac{H}{n}\right)^{\frac{r}{2}}2^{\frac{r}{2}M}$$

and from (27) we get that their sum is then

$$\left(\frac{H}{n2^\ell}\right)^{\frac{r}{2}}\sum_{J=1}^{M}2^{Jr} + \left(\frac{H}{n}\right)^{\frac{r}{2}}\sum_{J=1}^{M}2^{J\frac{r}{2}} \lesssim \left(\frac{H}{n2^\ell}\right)^{\frac{r}{2}}2^{rM}\vee\left(\frac{H}{n}\right)^{\frac{r}{2}}2^{\frac{r}{2}M} \lesssim \left(\frac{H}{n2^\ell}\right)^{\frac{rs}{2s+2}}\vee\left(\frac{H}{n}\right)^{\frac{rs}{2s+1}}.$$

Thus, it only remains to handle the third and fourth terms by choosing a suitable value for $\alpha$. Recalling that we are in the case $p > \frac{r}{s+1}$, we pick any $\frac{r}{s+1} < \alpha \leq p$; for instance, $\alpha := p$. Since $r - (s+1)p < 0$ we then have

$$\left(\frac{H}{n2^\ell}\right)^{\frac{r-p}{2}}\sum_{J=M}^{H}2^{J(r-(s+1)p)} \leq \left(\frac{H}{n2^\ell}\right)^{\frac{r-p}{2}}\frac{2^{M(r-(s+1)p)}}{1-2^{r-(s+1)p}} \asymp \left(\frac{H}{n2^\ell}\right)^{\frac{r-p}{2}}2^{M(r-(s+1)p)};$$

---

[1]To see why, recall that

$$\log m_J - J(2s+1) = \log\frac{n}{H} - (J-\ell)_+ - J(2s+1) + O(1)$$

from our setting of $m_J$. Finding the value of $J$ for which $\log\frac{n}{H} - (J-\ell)_+ - J(2s+1)$ cancels gives the claimed relation.

note that $\frac{1}{1-2^{r-(s+1)p}} > 0$ is a constant, depending only on $r, s, p$. Similarly, $r - (2s+1)p < 0$, and so

$$\left(\frac{H}{n}\right)^{\frac{r-p}{2}} \sum_{J=M}^{H} 2^{\frac{1}{2}J(r-(2s+1)p)} \leq \left(\frac{H}{n}\right)^{\frac{r-p}{2}} \frac{2^{\frac{1}{2}M(r-(2s+1)p)}}{1 - 2^{\frac{1}{2}(r-(2s+1)p)}} \asymp \left(\frac{H}{n}\right)^{\frac{r-p}{2}} 2^{\frac{1}{2}M(r-(2s+1)p)}.$$

From the setting of $M$ from (27), by a distinction of cases we again can bound their sum as

$$\left(\frac{H}{n2^\ell}\right)^{\frac{r-p}{2}} \sum_{J=M}^{H} 2^{J(r-(s+1)p)} + \left(\frac{H}{n}\right)^{\frac{r-p}{2}} \sum_{J=M}^{H} 2^{\frac{1}{2}J(r-(2s+1)p)} \lesssim \left(\frac{H}{n2^\ell}\right)^{\frac{rs}{2s+2}} \vee \left(\frac{H}{n}\right)^{\frac{rs}{2s+1}}.$$

Therefore, overall, in the case $p > \frac{r}{s+1}$ we have (bringing back the factor $H^{r/2}$ we had ignored earlier)

$$E_{bb} \lesssim H^{r/2} \left(\frac{H}{n2^\ell}\right)^{\frac{rs}{2s+2}} \vee H^{r/2} \left(\frac{H}{n}\right)^{\frac{rs}{2s+1}}. \tag{28}$$

- If $p \leq \frac{r}{s+1}$, we will choose $\alpha_J \geq p$ for all $J$. Under this constraint, we can use the monotonicity of $\ell_p$ norms (for every $x$, $\|x\|_p \leq \|x\|_q$ if $p \geq q$) to write

$$\begin{aligned}
E_{bb} &\lesssim \sum_{J=L}^{H} \frac{2^{J(r/2-1)}}{m_J^{(r-\alpha_J)/2}} \sum_{k \in \mathbb{Z}} |\beta_{J,k}|^{\alpha_J} \leq \sum_{J=L}^{H} \frac{2^{J(r/2-1)}}{m_J^{(r-\alpha_J)/2}} \left(\sum_{k \in \mathbb{Z}} |\beta_{J,k}|^p\right)^{\alpha_J/p} \\
&\lesssim \sum_{J=L}^{H} \frac{2^{J(r/2-1)}}{m_J^{(r-\alpha_J)/2}} 2^{-J\alpha_J(s+\frac{1}{2}-\frac{1}{p})} \qquad\qquad \text{(Fact A.3)} \\
&= \sum_{J=L}^{H} m_J^{-\frac{r-\alpha_J}{2}} 2^{J(\frac{r}{2}-1-\alpha_J(s+\frac{1}{2}-\frac{1}{p}))}.
\end{aligned}$$

As before, one can see that for there exists some $M$ such that the best choice is to set $\alpha_J = p$ for $J \leq M$ (as small as possible given our constraint $\alpha_J \geq p$). Moreover, proceeding as in the previous case,[2] we can see that this $M$ is such that

$$2^M \asymp \left(\frac{n2^\ell}{H}\right)^{\frac{1}{2(s-1/p)+2}} \wedge \left(\frac{n}{H}\right)^{\frac{1}{2(s-1/p)+1}}. \tag{29}$$

This part of the sum will then contribute

$$\begin{aligned}
\sum_{J=L}^{M} m_J^{-\frac{r-p}{2}} 2^{J(\frac{r}{2}-1-p(s+\frac{1}{2}-\frac{1}{p}))} &\asymp \left(\frac{H}{n2^\ell}\right)^{\frac{r-p}{2}} \sum_{J=L}^{M} 2^{J(r-1-p(s+1-\frac{1}{p}))} + \left(\frac{H}{n}\right)^{\frac{r-p}{2}} \sum_{J=L}^{M} 2^{J(\frac{r}{2}-1-p(s+\frac{1}{2}-\frac{1}{p}))} \\
&\asymp \left(\frac{H}{n2^\ell}\right)^{\frac{r-p}{2}} 2^{M(r-p(s+1))} + \left(\frac{H}{n}\right)^{\frac{r-p}{2}} 2^{M(\frac{r}{2}-1-p(s+\frac{1}{2}-\frac{1}{p}))} \\
&\asymp \left(\frac{H}{n2^\ell}\right)^{\frac{r(s-1/p+1/r)}{2(s-1/p)+2}} \vee \left(\frac{H}{n}\right)^{\frac{r(s-1/p+1/r)}{2(s-1/p)+1}}.
\end{aligned}$$

---

[2]That is, find the value $J$ solving (approximately) the equation $\log\frac{n}{H} - (J-\ell)_+ - J(2s+1-2/p) = 0$ (note that the LHS is again decreasing in $J$).

For $J > M$, we choose an arbitrary constant $\alpha \geq p$ such that $\alpha > \frac{r-1}{s+1-1/p}$ (so that $r - 1 - \alpha(s + 1 - 1/p) < 0$), and set $\alpha_J = \alpha$ for all $J > M$.[3] Observe that this implies $\alpha > \frac{r/2-1}{s+1/2-1/p}$. This part of the sum will then contribute at most

$$\sum_{J=M}^{H} m_J^{-\frac{r-\alpha}{2}} 2^{J(\frac{r}{2}-1-\alpha(s+\frac{1}{2}-\frac{1}{p}))} \asymp \left(\frac{H}{n2^\ell}\right)^{\frac{r-\alpha}{2}} \sum_{J=M}^{H} 2^{J(r-1-\alpha(s+1-\frac{1}{p}))} + \left(\frac{H}{n}\right)^{\frac{r-\alpha}{2}} \sum_{J=M}^{H} 2^{J(\frac{r}{2}-1-\alpha(s+\frac{1}{2}-\frac{1}{p}))}$$

$$\asymp \left(\frac{H}{n2^\ell}\right)^{\frac{r-\alpha}{2}} 2^{M(r-1-\alpha(s+1-\frac{1}{p}))} + \left(\frac{H}{n}\right)^{\frac{r-\alpha}{2}} 2^{M(\frac{r}{2}-1-\alpha(s+\frac{1}{2}-\frac{1}{p}))}$$

$$\asymp \left(\frac{H}{n2^\ell}\right)^{-\frac{r(s-1/p+1/r)}{2(s-1/p)+2}} \vee \left(\frac{H}{n}\right)^{\frac{r(s-1/p+1/r)}{2(s-1/p)+1}}$$

as well. Thus, overall, in the case $p \leq \frac{r}{s+1}$ we have (bringing back the factor $H^{r/2}$ we had ignored earlier)

$$E_{bb} \lesssim H^{r/2} \left(\frac{H}{n2^\ell}\right)^{\frac{r(s-1/p+1/r)}{2(s-1/p)+2}} \vee H^{r/2} \left(\frac{H}{n}\right)^{\frac{r(s-1/p+1/r)}{2(s-1/p)+1}}. \tag{30}$$

**The term $E_{sb}$.** To handle the term $E_{sb}$, we will rely on the fact that, for any $r \geq p$, we have the inclusion $\mathcal{B}(p,q,s) \subseteq \mathcal{B}(r,q,s')$, for $s' = s - \left(\frac{1}{p} - \frac{1}{r}\right)$. This will let us use Fact A.3 on $\sum_{k\in\mathbb{Z}} |\beta_{J,k}|^r$:

$$E_{sb} \lesssim H^{r/2} \sum_{J=L}^{H} 2^{J(\frac{r}{2}-1)} \sum_{k\in\mathbb{Z}} \mathbb{E}\left[|\beta_{J,k}|^r \mathbb{1}_{k\in\mathcal{I}_J^b\setminus\widehat{\mathcal{I}}_J}\right] \tag{Fact A.7}$$

$$\lesssim H^{r/2} \sum_{J=L}^{H} 2^{J(\frac{r}{2}-1)} \sum_{k\in\mathbb{Z}} |\beta_{J,k}|^r \Pr\left(|\widehat{\beta}_{J,k} - \beta_{J,k}| > \kappa t_J\right)$$

$$\lesssim H^{r/2} \sum_{J=L}^{H} 2^{J(\frac{r}{2}-1)} \sum_{k\in\mathbb{Z}} |\beta_{J,k}|^r 2^{-\kappa J} \tag{Lemma C.1}$$

$$\lesssim H^{r/2} \sum_{J=L}^{H} 2^{J(\frac{r}{2}-1-\kappa)} 2^{-Jr(s'+\frac{1}{2}-\frac{1}{r})} \tag{Fact A.3}$$

$$= H^{r/2} \sum_{J=L}^{H} 2^{-J(rs'+\kappa)} \leq H^{r/2} \frac{2^{-L(rs'+\kappa)}}{1 - 2^{-(rs'+\kappa)}}$$

$$\lesssim H^{r/2} 2^{-Lr(N+1)} \lesssim H^{r/2}(n2^\ell)^{-\frac{r(N+1)}{2(N+1)+2}} \vee H^{r/2} n^{-\frac{r(N+1)}{2(N+1)+1}} \leq H^{r/2}(n2^\ell)^{-\frac{rs}{2s+2}} \vee H^{r/2} n^{-\frac{rs}{2s+1}} \tag{31}$$

where for the third-to-last inequality we relied on our choice of $\kappa \geq r(N+1)$, and for the second-to-last, on our setting of $L$.

**The term $E_{ss}$.** Finally, we bound the last error term for details$(f)$, $E_{ss}$. In view of proceeding as for $E_{bb}$, for any nonnegative sequence $(\alpha_J)_J$ with $0 \leq \alpha_J \leq r$, we can write

$$E_{ss} \lesssim H^{r/2} \sum_{J=L}^{H} 2^{J(\frac{r}{2}-1)} \sum_{k\in\mathbb{Z}} \mathbb{E}\left[|\beta_{J,k}|^r \mathbb{1}_{k\in\mathcal{I}_J^s\setminus\widehat{\mathcal{I}}_J}\right] \tag{Fact A.7}$$

---

[3]It will be important later, when bounding $E_{ss}$, to note that $\frac{r-1}{s+1-1/p} < r$, and thus one can also enforce $\alpha \leq r$.

$$\leq H^{r/2} \sum_{J=L}^{H} 2^{J(\frac{r}{2}-1)} \sum_{k\in\mathbb{Z}} |\beta_{J,k}|^r \mathbb{1}_{k\in\mathcal{I}_J^s}$$

$$\leq H^{r/2} \sum_{J=L}^{H} 2^{J(\frac{r}{2}-1)} \sum_{k\in\mathbb{Z}} |\beta_{J,k}|^{\alpha_J} (\kappa t_J/2)^{r-\alpha_J} \mathbb{1}_{k\in\mathcal{I}_J^s}$$

$$\leq H^{r/2}(\kappa/2)^r \sum_{J=L}^{H} 2^{J(\frac{r}{2}-1)} J^{\frac{r-\alpha_J}{2}} m_J^{-(r-\alpha_J)/2} \sum_{k\in\mathbb{Z}} |\beta_{J,k}|^{\alpha_J}$$

$$\leq (\kappa/2)^r H^r \sum_{J=L}^{H} \frac{2^{J(r/2-1)}}{m_J^{(r-\alpha_J)/2}} \sum_{k\in\mathbb{Z}} |\beta_{J,k}|^{\alpha_J}$$

which is, except for the extra factor of $(\kappa/2)^r H^{\frac{r}{2}}$, exactly the same expression as (26). We can thus continue the analysis of $E_{ss}$ the same way as we did $E_{bb}$, noting that since $r \geq p$ all the choices for $\alpha_J$ in that analysis are still possible; leading to the bound:

$$E_{ss} \lesssim \begin{cases} H^r \cdot \left( \left(\frac{H}{n2^\ell}\right)^{\frac{rs}{2s+2}} \vee \left(\frac{H}{n}\right)^{\frac{rs}{2s+1}} \right) & p > \frac{r}{s+1} \\ H^r \cdot \left( \left(\frac{H}{n2^\ell}\right)^{\frac{r(s-1/p+1/r)}{2(s-1/p)+2}} \vee \left(\frac{H}{n}\right)^{\frac{r(s-1/p+1/r)}{2(s-1/p)+1}} \right) & p \leq \frac{r}{s+1}. \end{cases} \tag{32}$$

### C.3.3  Total error

Defining, for $r \geq 1$, $s \geq 0$, and $p \geq 1$, the quantities

$$\nu(r,p,s) := \frac{rs}{2s+2}\mathbb{1}_{p>\frac{r}{s+1}} + \frac{r(s-1/p+1/r)}{2(s-1/p)+2}\mathbb{1}_{p\leq\frac{r}{s+1}}$$

and

$$\mu(r,p,s) := \frac{rs}{2s+1}\mathbb{1}_{p>\frac{r}{s+1}} + \frac{r(s-1/p+1/r)}{2(s-1/p)+1}\mathbb{1}_{p\leq\frac{r}{s+1}}$$

we can gather all the error terms from Eqs. (23), (24), (25), (28), (30), (31) and (32), to get

$$\mathbb{E}\left[\left\|f-\widehat{f}\right\|_r^r\right] \lesssim H^\kappa \left( (n2^\ell)^{-\frac{r(s-1/p+1/r)}{2}} \vee n^{-r(s-1/p+1/r)} + (n2^\ell)^{-\frac{rs}{2s+2}} \vee n^{-\frac{rs}{2s+1}} + (n2^\ell)^{-\nu(r,p,s)} \vee n^{-\mu(r,p,s)} \right)$$

where $\kappa = \kappa(s,r,p)$ is a constant obtained for simplicity by taking the maximum of the exponent of $H$ in the previous bounds. To simplify this expression, we observe that the following holds for all $p, r, s \geq 1$:

- $\nu(r,p,s) \leq \frac{r(s-1/p+1/r)}{2}$

- $\nu(r,p,s) \leq \frac{rs}{2s+2}$

- $\mu(r,p,s) \leq r\left(s - \frac{1}{p} + \frac{1}{r}\right)$

- $\mu(r,p,s) > \frac{rs}{2s+1}$ if, and only if, $r \in ((s+1)p, (2s+1)p)$ (and of course $\mu(r,p,s) = \frac{rs}{2s+1}$ if $r \leq (s+1)p$)

(this follows from somewhat tedious algebraic manipulations and distinctions of cases). Given the above, we finally get the following bound (where we loosened the bound on the exponent of $H$ to make the result simpler to state):

$$\mathbb{E}\left[\left\|f-\widehat{f}\right\|_r^r\right] \lesssim H^\kappa \left( (n2^\ell)^{-\nu(r,p,s)} \vee n^{-\mu(r,p,s)} \vee n^{-\frac{rs}{2s+1}} \right)$$

$$= \log^{\kappa} n \cdot \begin{cases} (n2^{\ell})^{-\frac{rs}{2s+2}} \vee n^{-\frac{rs}{2s+1}} & \text{if } r < (s+1)p \\ (n2^{\ell})^{-\frac{r(s-1/p+1/r)}{2(s-1/p)+2}} \vee n^{-\frac{rs}{2s+1}} & \text{if } (s+1)p \le r < (2s+1)p \\ (n2^{\ell})^{-\frac{r(s-1/p+1/r)}{2(s-1/p)+2}} \vee n^{-\frac{r(s-1/p+1/r)}{2(s-1/p)+1}} & \text{if } r \ge (2s+1)p \end{cases} \quad (33)$$

which proves Theorem 1.3 in the main paper. $\qquad\square$

# D  Lower Bounds

Our lower bound construction will depend on whether $r < (s+1)p$ or $r \geq (s+1)p$. Before delving into these cases, we first (a) recall the result from [1] that we will use to upperbound average discrepancy; (b) discuss how the consideration of binary hypothesis testing problem gives a lower bound on average discrepancy.

## D.1  Upper bound on average discrepancy

Consider the following assumptions on $\mathcal{P} = \{\mathbf{p}_z : z \in \{-1,1\}^d\}$, where $\mathbf{p}_z$'s are probability distributions on $[0,1]$.

**Assumption D.1** (Densities exist). *For every $z \in \{-1,1\}^d$ and $i \in [\![d]\!]$, there exist functions $\phi_{z,i} : [0,1] \to \mathbb{R}$ such that $\mathbb{E}_{\mathbf{p}_z}[\phi_{z,i}^2] = 1$ and*

$$\frac{d\mathbf{p}_{z \oplus i}}{d\mathbf{p}_z} = 1 + \alpha \phi_{z,i}$$

*where $\alpha \in \mathbb{R}$ is a fixed constant independent of $z, i$.*

**Assumption D.2** (Orthonormality). *For all $z \in \{-1,1\}^d$ and $i,j \in [\![d]\!]$, $\mathbb{E}_{\mathbf{p}_z}[\phi_{z,i}\phi_{z,j}] = \mathbb{1}_{\{i=j\}}$.*

**Theorem D.3** (COROLLARY 2 in [1]). *Suppose $\mathcal{P}$ satisfies Assumptions D.1 and D.2. For some $\tau \in (0, 1/2]$, let $\pi$ be a prior on $Z \in \{-1,1\}^d$ defined as $Z_i \sim \mathtt{Rademacher}(\tau)$ independently for each $i \in [\![d]\!]$. For $Z \sim \pi$, let $X_1, \ldots, X_n$ be i.i.d. samples from $\mathbf{p}_Z$. Then, for any interactive protocol generating $\ell$-bit messages $Y_1, \ldots, Y_n$, we have*

$$\left( \frac{1}{d} \sum_{i=1}^d \mathrm{d_{TV}}\left( \mathbf{p}_{-i}^{Y^n}, \mathbf{p}_{+i}^{Y^n} \right) \right)^2 \leq \frac{7}{d} n \alpha^2 2^\ell.$$

## D.2  Lower bound on average discrepancy

For $Z \sim \pi$, let $X_1, \ldots, X_n$ be i.i.d. samples from $\mathbf{p}_Z$ distributed across $n$ players, and let $Y_1, \ldots, Y_n$ be $\ell$-bit messages sent by the players (possibly interactively) to the referee. Based on the $\ell$-bit messages $Y_1, \ldots, Y_n$, suppose the referee outputs an estimate $\hat{Z} = (\hat{Z}_1, \ldots, \hat{Z}_d)$ of $Z = (Z_1, \ldots, Z_d)$. Then, an upper bound on $\sum_{i=1}^d \Pr\left\{ \hat{Z}_i \neq Z_i \right\}$ gives a lower bound on $\sum_{i=1}^d \mathrm{D}\left( \mathbf{p}_{-i}^{Y^n} \| \mathbf{p}_{+i}^{Y^n} \right)$. To see this, note that, for a given $i \in [\![d]\!]$,

$$
\begin{aligned}
\Pr\left\{ \hat{Z}_i \neq Z_i \right\} &= \Pr\left\{ \hat{Z}_i = -1 | Z_i = 1 \right\} \Pr\left\{ Z_i = 1 \right\} + \Pr\left\{ \hat{Z}_i = 1 | Z_i = -1 \right\} \Pr\left\{ Z_i = -1 \right\} \\
&= \tau \left( 1 - \Pr\left\{ \hat{Z}_i = 1 | Z_i = 1 \right\} \right) + (1 - \tau) \Pr\left\{ \hat{Z}_i = 1 | Z_i = -1 \right\} \\
&\geq \tau \left( 1 - \Pr\left\{ \hat{Z}_i = 1 | Z_i = 1 \right\} \right) + \tau \Pr\left\{ \hat{Z}_i = 1 | Z_i = -1 \right\} \\
&\qquad\qquad\qquad\qquad\qquad\qquad\qquad\qquad \text{(since } (1 - \tau) \geq \tau \text{ for } \tau \leq 1/2) \\
&= \tau \left( 1 - \left( \Pr\left\{ \hat{Z}_i = 1 | Z_i = 1 \right\} - \Pr\left\{ \hat{Z}_i = 1 | Z_i = -1 \right\} \right) \right) \\
&\geq \tau \left( 1 - \mathrm{d_{TV}}\left( \mathbf{p}_{+i}^{Y^n}, \mathbf{p}_{-i}^{Y^n} \right) \right).
\end{aligned}
$$

Thus,

$$\sum_{i=1}^d \Pr\left\{ \hat{Z}_i \neq Z_i \right\} \geq \tau \left( d - \sum_{i=1}^d \mathrm{d_{TV}}\left( \mathbf{p}_{+i}^{Y^n}, \mathbf{p}_{-i}^{Y^n} \right) \right)$$

which gives

$$\frac{1}{d}\sum_{i=1}^{d}\mathrm{d_{TV}}\left(\mathbf{p}_{+i}^{Y^n},\mathbf{p}_{-i}^{Y^n}\right) \geq 1 - \frac{1}{d\tau}\sum_{i=1}^{d}\Pr\left\{\hat{Z}_i \neq Z_i\right\}. \tag{34}$$

In conclusion, to get a lower bound on average discrepancy, it suffices to upperbound $\sum_{i=1}^{d}\Pr\left\{\hat{Z}_i \neq Z_i\right\}$ for an estimator $\hat{Z}$ of $Z$.

### D.3 Lower bound on $\mathcal{L}_r^*(n,\ell,p,q,s)$ for $r < (s+1)p$

**Construction.** The family of distributions $\mathcal{P}_1$ that we will use to derive lower bound when $r < (s+1)p$ has also been used in deriving lower bounds in the unconstrained setting [4, 5] and in the LDP setting [2].

Let $g_0$ be a density function (see [5, p.157]) such that

1. $\mathrm{supp}(g_0) \subseteq [0,1]$;

2. $\|g_0\|_{pqs} \leq 1/2$;

3. $g_0 \equiv c_0 > 0$ on some interval $[a,b] \subseteq [0,1]$.

In what follows, $j$ is a free parameter that will be suitably chosen later in the proof. Let $\psi_{j,k}$ be defined as $\psi_{j,k}(x) = 2^{j/2}\psi(2^j x - k)$, where $\psi$ is the mother wavelet used to define $\|\cdot\|_{pqs}$ (see Section A). It is a fact that $\int \psi_{j,k}(x)dx = 0$ for every $j, k$ [5].

For a given $z \in \{-1,1\}^d$, define

$$f_z := g_0 + \gamma \sum_{k \in \mathcal{I}_j} z_k \psi_{j,k} \tag{35}$$

where

- $\mathcal{I}_j$ is the set of indices $k \in \mathbb{Z}$ such that

  i. $\mathrm{supp}(\psi_{j,k}) \subseteq [a,b]$ for every $k \in \mathcal{I}_j$;

  ii. for $k, k' \in \mathcal{I}_j$, $k \neq k'$, $\psi_{j,k}$ and $\psi_{jk'}$ have disjoint support;

  iii. $d := |\mathcal{I}_j| = C2^j$, for a constant $C$. Here on, we will assume for simplicity that $d = 2^j$.

- $\gamma$ is chosen such that

  i. $f_z(x) \geq c_0/2$ for every $x \in [a,b]$; this condition is satisfied if $c_0 - \gamma 2^{j/2}\|\psi\|_\infty \geq c_0/2$, i.e., $\gamma \leq (c_0/2\|\psi\|_\infty)2^{-j/2}$.

  ii. $\|f_z\|_{pqs} \leq 1$; since $\|f_z\|_{pqs} \leq \|g_0\|_{pqs} + \gamma\|\psi_{j,k}\|_{pqs} \leq 1/2 + \gamma C2^{j/p}2^{j(s+1/2-1/p)}$ (see pg. 160 in [5]), we get that $\|f_z\|_{pqs} \leq 1$ if $\gamma \leq (1/2C)2^{-j(s+1/2)}$.

  Since $s > 1/p > 0$, we get that for $j$ large enough, if $\gamma$ satisfies condition (ii), it automatically satisfies condition (i). Thus, we choose $\gamma = C2^{-j(s+1/2)}$ for some constant $C$.

Finally, we define the family of distributions as

$$\mathcal{P}_1 = \left\{\mathbf{p}_z : \mathbf{p}_z \text{ has density } f_z = g_0 + \gamma \sum_{k \in \mathcal{I}_j} z_k \psi_{j,k}, \ z \in \{-1,1\}^d\right\}. \tag{36}$$

**Prior on $Z$.** We assume a uniform prior on $Z \in \{-1, 1\}^d$, *i.e.*, $Z_i \sim \texttt{Rademacher}(1/2)$ independently for each $i \in [\![d]\!]$.

**Upper bound on average discrepancy.** To upperbound average discrepancy, we verify that $\mathcal{P}_1$ satisfies the three assumptions described in Section D.1, and then use Theorem D.3. For any $z \in \{-1, 1\}^d$, $k \in [\![d]\!]$, we have

$$\frac{d\mathbf{p}_{z \oplus k}}{d\mathbf{p}_z}(x) = 1 - \frac{2\gamma z_k \psi_{j,k}(x)}{c_0 + \gamma z_k \psi_{j,k}(x)}.$$

Since $\mathrm{supp}(\psi_{j,k}) \cap \mathrm{supp}(\psi_{j,k'})$ is empty for $k \neq k'$, it follows that Assumptions D.1 and D.2 hold. We now compute an upper bound on $\alpha^2 := \mathbb{E}_{\mathbf{p}_z} \left[ \left( \frac{\gamma z_k \psi_{j,k}(X)}{c_0 + \gamma z_k \psi_{j,k}(X)} \right)^2 \right]$.

$$
\begin{aligned}
\mathbb{E}_{\mathbf{p}_z} \left[ \left( \frac{2\gamma z_k \psi_{j,k}(X)}{c_0 + \gamma z_k \psi_{j,k}(X)} \right)^2 \right] &= 4\gamma^2 \int_{\mathrm{supp}(\psi_{j,k})} \frac{\psi_{j,k}(x)^2 (c_0 + \gamma z_k \psi_{j,k}(x))}{(c_0 + \gamma z_k \psi_{j,k}(x))^2} dx \\
&= 4\gamma^2 \int_{\mathrm{supp}(\psi_{j,k})} \frac{\psi_{j,k}(x)^2}{c_0 + \gamma z_k \psi_{j,k}(x)} dx \\
&\leq 2\gamma^2 c_0 \int_{\mathrm{supp}(\psi_{j,k})} \psi_{j,k}(x)^2 dx && (\text{as } c_0 + \gamma z_k \psi_{j,k}(x) \geq c_0/2) \\
&\leq 2\gamma^2 c_0 \times (2^{j/2} \|\psi\|_\infty)^2 \times \mathrm{length}(\mathrm{supp}(\psi_{j,k})) \\
&\leq 2\gamma^2 c_0 \times (2^{j/2} \|\psi\|_\infty)^2 \times \frac{C''}{2^j} && (\text{for a constant } C'' > 0) \\
&= C' \gamma^2 && (\text{for a constant } C' > 0) \\
&= C 2^{-j(2s+1)}. && (\text{for a constant } C > 0)
\end{aligned}
$$

Thus, using Theorem D.3, we get

$$\left( \frac{1}{2^j} \sum_{k \in \mathcal{I}_j} d_{\mathrm{TV}} \left( \mathbf{p}_{-k}^{Y^n}, \mathbf{p}_{+k}^{Y^n} \right) \right)^2 \lesssim (n2^\ell) 2^{-2j(s+1)}. \tag{37}$$

**Lower bound on average discrepancy.** To lower bound the average discrepancy, we will use the idea described in Section D.2. Consider a communication-constrained density estimation algorithm (possibly interactive) that outputs $\hat{f}$ satisfying $\sup_{f \in \mathcal{B}(p,q,s)} \mathbb{E}_f \left[ \|\hat{f} - f\|_r^r \right] \leq \varepsilon^r$. Using this density estimator, we estimate $\hat{Z}$ as

$$\hat{Z} = \underset{z}{\mathrm{argmin}} \left\| f_z - \hat{f} \right\|_r.$$

Then

$$\mathbb{E}_{\mathbf{p}_z} \left[ \|f_z - f_{\hat{Z}}\|_r^r \right] \leq 2^{r-1} \left( \mathbb{E}_{\mathbf{p}_z} \left[ \left\| f_z - \hat{f} \right\|_r^r \right] + \mathbb{E}_{\mathbf{p}_z} \left[ \|f_z - f_{\hat{Z}}\|_r^r \right] \right) \leq 2^r \varepsilon^r. \tag{38}$$

Now, for $z \neq z'$, we have

$$
\begin{aligned}
\|f_z - f_{z'}\|_r^r &= \int_0^1 |f_z(x) - f_{z'}(x)|^r dx \\
&= \gamma^r \int_0^1 \left| \sum_{k \in \mathcal{I}_j} \psi_{j,k}(x) \mathbb{1}_{\{z_k \neq z'_k\}} \right|^r dx
\end{aligned}
$$

$$= \gamma^r \int_0^1 \sum_{k \in \mathcal{I}_j} |\psi_{j,k}(x)|^r \mathbb{1}_{\{z_k \neq z'_k\}} dx \qquad \text{(since the } \psi_{j,k}\text{'s have disjoint supports)}$$

$$= \gamma^r \sum_{k \in \mathcal{I}_j} \int_{\text{supp}(\psi_{j,k})} |\psi_{j,k}(x)|^r \mathbb{1}_{\{z_k \neq z'_k\}} dx$$

$$= \gamma^r (2^{j/2} \|\psi\|_\infty)^r \frac{C''}{2^j} \sum_{k \in \mathcal{I}_j} \mathbb{1}_{\{z_k \neq z'_k\}} \qquad \text{(for a constant } C'' > 0)$$

$$= C' \gamma^r 2^{j(r/2-1)} \sum_{k \in \mathcal{I}_j} \mathbb{1}_{\{z_k \neq z'_k\}} \qquad \text{(for a constant } C' > 0)$$

$$= C 2^{-j(rs+1)} \sum_{k \in \mathcal{I}_j} \mathbb{1}_{\{z_k \neq z'_k\}}. \qquad \text{(for a constant } C > 0)$$

which gives that, for an estimator $\hat{Z}$,

$$\mathbb{E}_{\mathbf{p}_z} \left[ \|f_z - f_{\hat{Z}}\|_r^r \right] = C 2^{-j(rs+1)} \sum_{k \in \mathcal{I}_j} \Pr \left\{ Z_k \neq \hat{Z}_k \right\}.$$

Combining this with (38), we get

$$\sum_{k \in \mathcal{I}_j} \Pr \left\{ \hat{Z}_k \neq Z_k \right\} \lesssim \varepsilon^r 2^{j(rs+1)}. \tag{39}$$

Thus, substituting $d = 2^j$ and $\tau = 1/2$ in (34) (and ignoring multiplicative constants), we get

$$\frac{1}{2^j} \sum_{k \in \mathcal{I}_j} d_{\text{TV}} \left( \mathbf{p}_{-k}^{Y^n}, \mathbf{p}_{+k}^{Y^n} \right) \gtrsim 1 - \frac{2}{2^j} \varepsilon^r 2^{j(rs+1)} \simeq 1 - \varepsilon^r 2^{jrs}.$$

Now, observe that $j$ is a free parameter that we can choose. If we choose $j$ such that

$$\varepsilon^r 2^{jrs} \simeq 1 \tag{40}$$

then we get

$$\frac{1}{2^j} \sum_{k \in \mathcal{I}_j} d_{\text{TV}} \left( \mathbf{p}_{-k}^{Y^n}, \mathbf{p}_{+k}^{Y^n} \right) \gtrsim 1$$

or

$$\left( \frac{1}{2^j} \sum_{k \in \mathcal{I}_j} d_{\text{TV}} \left( \mathbf{p}_{-k}^{Y^n}, \mathbf{p}_{+k}^{Y^n} \right) \right)^2 \gtrsim 1 \tag{41}$$

**Putting things together.** From (37) and (41), we get that, for $j$ satisfying $\varepsilon^r 2^{jrs} \simeq 1$,

$$1 \lesssim (n 2^\ell) 2^{-2j(s+1)}.$$

which gives

$$2^j \lesssim (n 2^\ell)^{\frac{1}{2s+2}}.$$

Using $\varepsilon^r 2^{jrs} \simeq 1$, we finally get

$$\varepsilon^r \gtrsim (n 2^\ell)^{-\frac{rs}{2s+2}}$$

which is our desired lower bound.

## D.4 Lower bound on $\mathcal{L}_r^*(n, \ell, p, q, s)$ for $r \geq (s+1)p$

**Construction.** The family of distributions $\mathcal{P}_2$ that we will use to derive lower bound when $r \geq (s+1)p$ is not exactly the same as that in the unconstrained and in the LDP setting [4, 5, 2]; but, combined with the prior that we will choose on $Z$, it will essentially mimic that.

Let $g_0, \psi_{j,k}, \mathcal{I}_j$ be as in Section D.3. For a given $z \in \{-1, 1\}^d$ (where $d := |\mathcal{I}_j|$), define

$$f_z := g_0 + \gamma \sum_{k \in \mathcal{I}_j} (1 + z_k)\psi_{j,k}. \tag{42}$$

where we will choose $\gamma$ after we describe the prior on $Z$. Finally, we define the family of distributions as

$$\mathcal{P}_2 = \left\{ \mathbf{p}_z : \mathbf{p}_z \text{ has density } f_z = g_0 + \gamma \sum_{k \in \mathcal{I}_j} (1 + z_k)\psi_{j,k}, \ z \in \{-1, 1\}^d \right\}. \tag{43}$$

**Prior on $Z$.** We assume a "sparse" prior on $Z \in \{-1, 1\}^d$, defined as $Z_k \sim \texttt{Rademacher}(1/d)$ independently for each $k \in [\![d]\!]$. We call it "sparse" because, with high probability, for $Z = (Z_1, \ldots, Z_d)$ sampled from this prior, the number of indices $k$ with $Z_k = 1$ will be small (we will quantify this soon). Now, since $f_Z = g_0 + \gamma \sum_{k \in \mathcal{I}_j} (1 + Z_k)\psi_{j,k}$, this means that with high probability $1 + Z_k = 0$ for a large number of $k$'s, and thus there will be only a few "bumps" in $f_Z$.

**Choosing $\gamma$.** Define $\mathcal{G} \subset \{-1, 1\}^d$ as

$$\mathcal{G} := \left\{ z \in \{-1, 1\}^d : \sum_{k=1}^d \mathbb{1}_{\{z_k = 1\}} \leq 2j \right\}.$$

Then, by Bernstein's inequality

$$\Pr\{Z \in \mathcal{G}\} \geq 1 - 4 \cdot 2^{-2j}. \tag{44}$$

We will choose $\gamma$ such that

    i. $f_z(x) \geq c_0/2$ for every $x \in [a, b]$; as seen in Section D.3, this condition is satisfied if $\gamma \leq (c_0/2\|\psi\|_\infty)2^{-j/2}$.

    ii. $\|f_z\|_{pqs} \leq 1$ for every $z \in \mathcal{G}$; argument similar to that in Section D.3 gives that $\|f_z\|_{pqs} \leq 1$ for $z \in \mathcal{G}$ if $\gamma \lesssim 2^{-j(s+1/2-1/p)}j^{-1/p}$.

Since $s > 1/p$, we get that for $j$ large enough ($j$ is a free parameter that we choose later), if $\gamma$ satisfies condition (ii), it automatically satisfies condition (i). Thus, we choose $\gamma = C2^{-j(s+1/2-1/p)}j^{-1/p}$ for some constant $C$.

Note that, for $z \notin \mathcal{G}$, this choice of $\gamma$ still results in $f_z$ being a density function (since $\int \psi_{j,k}(x)dx = 0$), but it may be the case that $\|f_z\|_{spq} > 1$.

**Upper bound on average discrepancy.** To upperbound average discrepancy, we verify that $\mathcal{P}_2$ satisfies the three assumptions described in Section D.1. For any $z \in \{-1,1\}^d$, $k \in [\![d]\!]$, we have

$$
\frac{dp_{z \oplus k}}{dp_z}(x) = \frac{g_0 + \gamma \sum_{k \in I_j}(1 - z_k)\psi_{j,k}(x)}{g_0 + \gamma \sum_{k \in I_j}(1 + z_k)\psi_{j,k}(x)}
$$

$$
= 1 - \frac{2\gamma z_k \psi_{j,k}(x)}{c_0 + \gamma \sum_{k \in I_j} z_k \psi_{j,k}(x)}
$$

which is same as what we had in Section D.3. Similar arguments lead to the conclusion that Assumptions D.1 and D.2 are satisfied. Moreover, an upper bound on $\alpha^2 := \mathbb{E}_{\mathbf{p}_z}\left[\left(\frac{2\gamma z_i \psi_{j,k}(X)}{c_0 + \gamma z_i \psi_{j,k}(X)}\right)^2\right]$ follows similarly (with different value of $\gamma$), and we get that

$$
\alpha^2 \leq C 2^{-2j(s+1/2-1/p)} j^{-2/p}. \tag{for a constant $C > 0$}
$$

Thus, using Theorem D.3, we get

$$
\left(\frac{1}{2^j} \sum_{k \in \mathcal{I}_j} d_{\mathrm{TV}}\left(\mathbf{p}_{-k}^{Y^n}, \mathbf{p}_{+k}^{Y^n}\right)\right)^2 \lesssim (n2^\ell) 2^{-2j(s+1-1/p)} j^{-2/p}. \tag{45}
$$

**Lower bound on average discrepancy.** To lowerbound average discrepancy, we proceed as in Section D.3. Consider a communication-constrained density estimation algorithm (possibly interactive) that outputs $\hat{f}$ satisfying $\sup_{f \in \mathcal{B}(p,q,s)} \mathbb{E}_f\left[\|\hat{f} - f\|_r^r\right] \leq \varepsilon^r$. Using this density estimator, we estimate $\hat{Z}$ as

$$
\hat{Z} = \underset{z}{\mathrm{argmin}} \left\|f_z - \hat{f}\right\|_r.
$$

Then, for $z \in \mathcal{G}$,

$$
\mathbb{E}_{\mathbf{p}_z}\left[\|f_z - f_{\hat{Z}}\|_r^r\right] \leq 2^r \varepsilon^r. \tag{46}
$$

This only holds for $z \in \mathcal{G}$ because the estimator's guarantee only holds if samples come from a density $f$ satisfying $\|f\|_{spq} \leq 1$. Now, for $z \neq z'$, plugging in the value of $\gamma$ in the calculation done in Section D.3, we get

$$
\|f_z - f_{z'}\|_r^r = Cj^{-r/p} 2^{-j(r(s-1/p)+1)} \sum_{k \in \mathcal{I}_j} \mathbb{1}_{\left\{z_k \neq z_k'\right\}} \tag{for a constant $C > 0$}
$$

which gives that, for any estimator $\hat{Z}$,

$$
\mathbb{E}_{\mathbf{p}_z}\left[\|f_z - f_{\hat{Z}}\|_r^r\right] = Cj^{-r/p} 2^{-j(r(s-1/p)+1)} \sum_{k \in \mathcal{I}_j} \Pr\left\{\hat{Z}_k \neq Z_k\right\}.
$$

Combining this with (46), we get that

$$
\sum_{k \in \mathcal{I}_j} \Pr\left\{Z_k \neq \hat{Z}_k, Z \in \mathcal{G}\right\} \lesssim \varepsilon^r j^{r/p} 2^{j(r(s-1/p)+1)}. \tag{47}
$$

Thus,

$$
\sum_{k \in \mathcal{I}_j} \Pr\left\{\hat{Z}_k \neq Z_k\right\} = \sum_{k \in \mathcal{I}_j} \Pr\left\{Z_k \neq \hat{Z}_k, Z \in \mathcal{G}\right\} + \sum_{k \in \mathcal{I}_j} \Pr\left\{Z_k \neq \hat{Z}_k, Z \notin \mathcal{G}\right\}
$$

$$\leq \sum_{k \in \mathcal{I}_j} \Pr\left\{ Z_k \neq \hat{Z}_k, Z \in \mathcal{G} \right\} + \left( \sum_{k \in \mathcal{I}_j} \Pr\left\{ \hat{Z}_k \neq Z_k | Z \notin \mathcal{G} \right\} \right) \Pr\left\{ Z \notin \mathcal{G} \right\}$$

$$\lesssim \varepsilon^r j^{r/p} 2^{j(r(s-1/p)+1)} + 2^j 2^{-2j}. \tag*{(using (47),(44))}$$

Thus, substituting $d = 2^j$ and $\tau = 1/d = 2^{-j}$ in (34) (and ignoring multiplicative constants), we get

$$\frac{1}{2^j} \sum_{k \in \mathcal{I}_j} \mathrm{d_{TV}}\left( \mathbf{p}_{-k}^{Y^n}, \mathbf{p}_{+k}^{Y^n} \right) \gtrsim 1 - \varepsilon^r j^{r/p} 2^{j(r(s-1/p)+1)} - 2^{-j}$$

$$\simeq 1 - \varepsilon^r j^{r/p} 2^{jr(s-1/p+1/r)}.$$

Choosing $j$ such that

$$\varepsilon^r 2^{jr(s-1/p+1/r)} j^{r/p} \simeq 1 \tag{48}$$

gives

$$\frac{1}{2^j} \sum_{k \in \mathcal{I}_j} \mathrm{d_{TV}}\left( \mathbf{p}_{-k}^{Y^n}, \mathbf{p}_{+k}^{Y^n} \right) \gtrsim 1.$$

or

$$\left( \frac{1}{2^j} \sum_{k \in \mathcal{I}_j} \mathrm{d_{TV}}\left( \mathbf{p}_{-k}^{Y^n}, \mathbf{p}_{+k}^{Y^n} \right) \right)^2 \gtrsim 1. \tag{49}$$

**Putting things together.** From (45) and (49), we get, for any $j$ satisfying $\varepsilon^r 2^{jr(s-1/p+1/r)} j^{r/p} \simeq 1$, that $1 \lesssim (n2^\ell) 2^{-2j(s+1-1/p)} j^{-2/p}$. This then yields

$$2^{2j(s+1-1/p)} j^{2/p} \lesssim n2^\ell. \tag{50}$$

To get a rough idea of the bound this will give, let us ignore $j^{2/p}$ to get,

$$2^j \lesssim \left( n2^\ell \right)^{\frac{1}{2(s+1-1/p)}}. \tag{51}$$

Now, since $\varepsilon^r 2^{jr(s-1/p+1/r)} j^{r/p} \simeq 1$, we get, roughly, (ignoring $j^{r/p}$)

$$2^j \simeq (1/\varepsilon)^{\frac{1}{s-1/p+1/r}}. \tag{52}$$

Combining (51), (52), we get that (up to logarithmic factors)

$$\varepsilon^r \gtrsim (n2^\ell)^{-\frac{r(s-1/p+1/r)}{2(s+1-1/p)}}$$

which is the desired bound, again up to logarithmic factors. We now show how a slightly more careful analysis lets us obtain the tight bound.

**Bringing in log factors.** From (50), we get that

$$2^j \lesssim \left( n2^\ell \right)^{\frac{1}{2(s+1-1/p)}} \left( \log(n2^\ell) \right)^{-\frac{2/p}{2(s+1-1/p)}}. \tag{53}$$

Now, since $\varepsilon^r 2^{jr(s-1/p+1/r)} j^{r/p} \simeq 1$, we get

$$2^j \simeq (1/\varepsilon)^{\frac{1}{s-1/p+1/r}} \left( \log(1/\varepsilon) \right)^{-\frac{1}{p(s-1/p+1/r)}}.$$

Substituting this in (53), we get

$$(1/\varepsilon)^{\frac{1}{s-1/p+1/r}} \left(\log(1/\varepsilon)\right)^{-\frac{1}{p(s-1/p+1/r)}} \lesssim \left(n2^\ell\right)^{\frac{1}{2(s+1-1/p)}} \left(\log(n2^\ell)\right)^{-\frac{2/p}{2(s+1-1/p)}}$$

or

$$1/\varepsilon \left(\log(1/\varepsilon)\right)^{-1/p} \lesssim \left(n2^\ell\right)^{\frac{s-1/p+1/r}{2(s+1-1/p)}} \left(\log(n2^\ell)\right)^{-(2/p)\frac{(s-1/p+1/r)}{2(s+1-1/p)}} .$$

This implies that

$$1/\varepsilon \lesssim \left(n2^\ell\right)^{\frac{s-1/p+1/r}{2(s+1-1/p)}} \left(\log(n2^\ell)\right)^{-(2/p)\frac{(s-1/p+1/r)}{2(s+1-1/p)}} \left(\log\left(\left(n2^\ell\right)^{\frac{s-1/p+1/r}{2(s+1-1/p)}} \left(\log(n2^\ell)\right)^{-(2/p)\frac{(s-1/p+1/r)}{2(s+1-1/p)}}\right)\right)^{1/p}$$

$$\simeq \left(n2^\ell\right)^{\frac{s-1/p+1/r}{2(s+1-1/p)}} \left(\log(n2^\ell)\right)^{-(2/p)\frac{(s-1/p+1/r)}{2(s+1-1/p)}+\frac{1}{p}}$$

$$= \left(n2^\ell\right)^{\frac{s-1/p+1/r}{2(s+1-1/p)}} \left(\log(n2^\ell)\right)^{-\frac{1-1/r}{p(s-1/p+1)}} .$$

Thus

$$\varepsilon^r \gtrsim \left(n2^\ell\right)^{\frac{r(s-1/p+1/r)}{2(s+1-1/p)}} \left(\log(n2^\ell)\right)^{-\frac{r-1}{p(s-1/p+1)}} . \tag{54}$$

## D.5 Concluding the proof of Theorem 1.1

Combining lower bounds from Sections D.3 and D.4 with lower bounds in the classical setting [4] (where the rate transition happens at $r = (2s+1)p$), we get Theorem 1.1. $\qquad\square$