# OpenReview forum: "Optimal Rates for Nonparametric Density Estimation under Communication Constraints"
_NeurIPS.cc/2021/Conference — NeurIPS 2021 Poster_

### Official Review · Reviewer_4wEN · 2021-07-06

**Rating:** 6
**Confidence:** 3

**Summary:**

The paper proposed a method for distributed density estimation with the unknown density from a Besov space. The estimation is based on local observations of independent samples drawn from a common distribution. The agents with local observations are connected via bit-constrained channels to a central unit ("referee"). The proposed method is based on a novel quantization strategy that exploits the properties of a wavelet expansion of the unknown density. The method is implemented as two algorithms, one which requires knowledge of Besov space parameter and another one which does not require precise Besov parameters.


**Limitations And Societal Impact:**

yes

**Main Review:**

There should be more motivation for why it is useful to consider the extension from Hoelder to Besov spaces. Which application domains (e.g. wireless sensor networks) involve densities which belong to Besov but not to Hoelder spaces? I also suggest adding numerical experiments to showcase the usefulness of your methods compared to state-of-the art.

I would like to have more discussion about the restrictiveness of the assumption of wavelet-induced sparsity as formulated in Claim 3.1. Also, i suggest renaming Claim 3.1 either as Assumption or as Corollary depending if this follows from other assumptions on the wavelet expansion or is an explicit assumption that you require to be fulfilled for your method.

The considered communication setting seems quite simple. The paper mentioned federated learning as an application domain for density estimation. It would be interesting to discuss how the proposed methods could be extended to fully decentralized settings where each agent is participating equally in the estimation. Such a setting has been considered for parametric estimation recently in

A. Jung, "Networked Exponential Families for Big Data Over Networks," in IEEE Access, vol. 8, pp. 202897-202909, 2020, doi: 10.1109/ACCESS.2020.3033817.


The clarity/precision of the presentation could be improved:

- a quick reminder about the defs for Hoelder and Besov spaces would be helpful for those who don't work with them on a daily basis.

- The role of the last paragraph in Section 1.2. is not very clear.

- in which sense is (4) to be understood ? pointwise (for each x) or w.r.t. some norm?

- ".. is an unbiased estimate of \alpha_{j,k} .." how is \alpha_{j,k} defined for j different from 0 ?

- "...s lies in a certain range." what is this range?

- "Our proposed estimators draws upon these classical estimators.."  how exactly does it draw upon them ?

- "As in the unconstrained setting ..."  i assume you mean for \ell -> infty ?

- include an outline paragraph in Section 1 where you give a plan of the paper and also make explicit how the different parts tie in together. E.g., What is the relation between Algorithm 1,2,3,4,5 ?

- be more precise about inputs to Algorithms. E.g., Algorithm 2 uses a parameter "H", "\ell" and "\phi_{H,k}(.)" which seem to be input parameter. Or do you consider only one single choice for those throughout the entire paper.

- ".. we form an approximately sufficient statistic ... can be represented using finite number of bits .." pls refer to the precise formula for this approximately sufficient statistic

- the relevance of Theorem 3.3. is not very clear. I also suggest to write down DISTRSIM algorithm explicitly as it is a main building block of your methods.



**Time Spent Reviewing:**

4

---

> ### Author Response · Authors · 2021-08-10
> **We thank the reviewer for their time; we respond to the points raised below.**
>
> - "There should be more motivation for why it is useful to consider the extension from Hoelder to Besov spaces. Which application domains (e.g. wireless sensor networks) involve densities which belong to Besov but not to Hoelder spaces?"
>
> In practical applications, it is generally difficult to a priori state with certainty the exact class to which the unknown density function belongs. Looking at richer classes corresponds to imposing fewer assumptions on the unknown density and letting the data “speak for itself” (in fact, this is one of the main reasons to prefer nonparametric models over parametric models), especially since adaptive algorithms are “hyperparameter-free.” In nonparametric theory, Besov classes are the most general function classes considered, and include more traditional classes like Hölder and Sobolev. From the point of view of a practitioner, our adaptive algorithm removes the need to make assumptions about the precise class to which the unknown density might belong, while still achieving near-optimal rates. In particular, our adaptive algorithm guarantees near-optimal rates for simpler classes without explicitly knowing them. On the other hand, algorithms designed for simpler classes (e.g., Hölder) do not extend to more complicated ones.
> It is also worth mentioning that, in the past, the choice to study Besov classes has led to interesting nonparametric estimation algorithms and fruitful insights (and for instance considerably advanced the understanding of adaptivity in nonparametric estimation). This is one of the reasons which has recently led researchers  (including us for the communication-constrained setting) to consider this class of functions in some information-constrained settings. In particular, many of the interesting phenomena that we uncover are not present in the simpler Holder class, and thus, focusing on this class may lead one to use a weaker class of algorithms.
>
> - "I also suggest adding numerical experiments to showcase the usefulness of your methods compared to state-of-the art."
>
> We agree that experimental evaluation and comparison seem worth exploring, especially as our algorithm does seem practical; and we view this as a great suggestion for follow-up work (see below). However, the current paper is theoretical in nature and meant as proposing a general way to quantize for density estimation.
> Regarding practical evaluation: we are in fact pursuing this as follow-up work, and are in the process of designing the experiment – looking for datasets; however, this is a significant undertaking, and we chose to separate the theoretical and practical aspects in the interest of time, space, and focus of the papers. We nonetheless thank the reviewer for their suggestion, as it will be interesting to see if the shortcomings of wavelet-based methods in comparison with kernel-based methods carry over to the communication-constrained setting as well.
>
> - "I would like to have more discussion about the restrictiveness of the assumption of wavelet-induced sparsity as formulated in Claim 3.1. Also, i suggest renaming Claim 3.1 either as Assumption or as Corollary depending if this follows from other assumptions on the wavelet expansion or is an explicit assumption that you require to be fulfilled for your method."
>
> Claim 3.1 itself is not an assumption, but a consequence of the assumption that the wavelet functions have compact support. Wavelet functions with compact support are known (e.g., Daubechies wavelet).  Since the choice of the wavelet function to be used is in the designer’s hands, the assumption of compact support is not restrictive.
>
> - "The considered communication setting seems quite simple. The paper mentioned federated learning as an application domain for density estimation. It would be interesting to discuss how the proposed methods could be extended to fully decentralized settings where each agent is participating equally in the estimation."
>
> Our distributed model is a natural setting to study the problem of designing quantization algorithms for general density classes. This can also be seen as a first and necessary step towards studying more general communication models (e.g., decentralized) for density estimation. We believe that algorithms for more general models will benefit from the insights obtained in our simple communication-constrained setting.
>
> - "The clarity/precision of the presentation could be improved:"
> "a quick reminder about the defs for Hoelder and Besov spaces would be helpful for those who don't work with them on a daily basis."
> We have defined Besov in the Appendix; to clarify things, we will add an explicit reference to it in the problem setup (Section 1.1), and will define Holder in Introduction or in Prior work.
>
> - "The role of the last paragraph in Section 1.2. is not very clear."
> The paragraph discusses our attempt at working with the natural basis (Fourier basis) for the Sobolev class. This serves to highlight the utility of wavelet bases in communication-constrained settings, and explain why we proceed with this basis instead of the more natural Fourier one (which is more natural, but doesn’t seem  to lead to optimal rates).
>
> - "in which sense is (4) to be understood ? pointwise (for each x) or w.r.t. some norm?"
> This is to be understood as convergence in the L2 sense; we will add a sentence to clarify this.
>
> - ".. is an unbiased estimate of \alpha_{j,k} .." how is \alpha_{j,k} defined for j different from 0 ?
> This is defined explicitly in the appendix; we will clarify this by providing an explicit reference to that definition in the main text. For the purposes of our discussion, we only need to know that \alpha_{j,k} is the coefficient of \phi_{j,k} in the wavelet expansion.
>
> - "...s lies in a certain range." what is this range?
> This is part of the discussion of the work of Donoho et al. (Line 166), and refers to their bound (s>1/N). We use the same one, and will clarify the language in that sentence accordingly.
>
> - "Our proposed estimators draws upon these classical estimators.." how exactly does it draw upon them ?
> We estimate wavelet coefficients as in the centralized setting. Moreover, as in the centralized setting, we use thresholding for adaptive estimation.
>
> - "As in the unconstrained setting ..." i assume you mean for \ell -> infty ?
> Indeed, this is what we mean (we used “unconstrained” for “absent any communication constraint”). Equivalently, in the unconstrained setting, the estimator can access samples X_1,...,X_n directly.
>
> - "include an outline paragraph in Section 1 where you give a plan of the paper and also make explicit how the different parts tie in together. E.g., What is the relation between Algorithm 1,2,3,4,5 ?"
> This is a good point: we will add a paragraph on organization at the end of Section 1.
>
> - "be more precise about inputs to Algorithms. E.g., Algorithm 2 uses a parameter "H", "\ell" and "\phi_{H,k}(.)" which seem to be input parameter. Or do you consider only one single choice for those throughout the entire paper."
> While we discussed this in the text of the paper (Lines 239-241, Lines 274-278), we will make the inputs of our algorithms explicit in the pseudocode. Other than that, another input is the player’s samples.
>
> - ".. we form an approximately sufficient statistic ... can be represented using finite number of bits .." pls refer to the precise formula for this approximately sufficient statistic
> Each player has nonzero coefficients, which she quantizes and can still obtain optimal rates up to constants (thus, “approximately sufficient statistic”) if \ell is a sufficiently large finite value. We quantify what this “sufficiently large” exactly means in the equation after Line 73.
>
> - "the relevance of Theorem 3.3. is not very clear. I also suggest to write down DISTRSIM algorithm explicitly as it is a main building block of your methods."
> One way to think of it is that we have reduced the problem to that of discrete distribution estimation, where DISTRSIM is known to give optimal rates. Theorem 3.3 just summarizes the guarantees of DISTRSIM and makes it clear how many samples can be simulated by n players.

---

> > ### Comment · Reviewer_4wEN · 2021-08-19
> > **Significance of Results**
> >
> > I thank the authors for carefully addressing my comments but my main concern is still the scope and significance of the presented results. I agree with the authors that it is conceptually appealing to use less restrictive modeling assumptions, or equivalently, allow for larger function spaces. However, there should be more discussion on the trade-offs between statistical performance (smaller minimax errors) and computational costs offered by the proposed method in contrast to existing methods. Can you provide an example for a function from a Besov space that cannot be well estimated by existing distributed estimation methods for simpler functions spaces? Also, how does the computational complexity of your method compare with existing methods? Does your method achieve minimax risk for Besov spaces and simpler spaces with no significant additional computational cost? Instead of a theoretical analysis, numerical experiments could be used for such a comparison.

---

> > > ### Author Response · Authors · 2021-08-20
> > > **Addressing the Reviewer's comments and questions**
> > >
> > > The question raised applies not only to our work in the communication-constrained setting, but to large literature on nonparametric estimation that has come up over the past three decades. By considering Besov spaces, statisticians have discovered new estimators which are not only minmax optimal but can adapt to the smoothness of the function class. While it is easy to provide examples of densities which are in Besov class or Sobolev class and in Holder class, but we think that the point is not to give such examples. After all, these are all models and none of them may actually be exact for the underlying data. In our view, the true value of richer model is to get new algorithms that apply to broader data. In any case, since our current paper is focusing on theory, we would like to point out that Besov spaces are standard* and results for Holder class alone are rather weak. We would like to reiterate that this discussion is not strictly applicable only to our setting, but is a general discussion about nonparametric estimation.
> > >
> > > About the computation complexity, indeed wavelets are computationally heavy (unless we use dedicated hardware) and developing more efficient algorithms using Kernel methods will be an interesting direction. However, in terms of theoretical computational complexity the proposed scheme is efficient.
> > >
> > > * To address the specific question on what types of functions 'belong to a Besov space yet cannot be well estimated by existing distributed estimation methods for simpler functions spaces,' we would like to point to many natural applications such as those arising from physical processes. In particular, most fluid dynamics applications, where the data measured follows a density satisfying a set of PDEs (Navier-Stokes, etc) will not be captured by those simpler function spaces; and indeed, these physical applications are in great part the very motivation for introducing Sobolev and Besov spaces in the first place. Note however that these are not specific to the communication-constrained setting, but arise in the centralized setting as well.

---

> > > > ### Comment · Reviewer_4wEN · 2021-08-20
> > > > **Changing Rating**
> > > >
> > > > Thx for the clarification. I have modified my rating accordingly.

---

### Official Review · Reviewer_SLwN · 2021-07-15

**Rating:** 7
**Confidence:** 3

**Summary:**

This problem of estimating a density, assumed to lie in a Besov class, under communication constraints in which each sample must be encoded using a fixed number $\ell$ of bits before computing the density estimate. Both the non-adaptive and adaptive cases are considered, with matching upper and lower bounds (up to polylogarithmic factors) obtained in both cases. The matching bounds also indicate that the rate is the same for interactive and non-interactive protocols. The proposed methods utilizes 3 main ingredients:
1. The fact that at most a constant number of wavelets at resolution J are non-zero on any interval of width $2^{-J}$.
2. A particular randomized scheme that quantizes the values of the non-zero wavelets at a sample point without introducing bias.
3. The DISTRSIM protocol proposed by [3], which maps a quantized sample to a smaller IID sample of higher resolution.

**Limitations And Societal Impact:**

The assumptions underlying the paper's results are fairly clearly stated. The results are fairly comprehensive for the problem being studied, although additional discussion of room for future work would be nice. I don't foresee any potential negative societal impact of this work.

**Main Review:**

Overall, this paper was quite well-written, and both the problem and its solution are well-motivated and interesting. My main criticism is that no experiments are performed. It would be interesting, for example, to know how large $n$ needs to be for the proposed method to outperform the "natural approach" (Lines 83-85). However, I think the paper makes a sufficiently strong theoretical contribution to be accepted even without experimental results.

Comments:
1) It's surprising to me that one can even estimate the density consistently as $n \to \infty$ if $\ell$ is fixed (using a non-interactive protocol). Fixing the number $\ell$ of bits from each sample, it seems like there would be some resolution $J$ below which it would not be possible to encode all of the possible bins into which the samples might fall. Certainly, I think the "natural approach" (Lines 83-85) would fail to converge for this reason. The trick to overcoming this seems to lie in the DISTRSIM protocol, since that produces arbitrarily many bits of each sample so long as $n$ is sufficiently large. It would be nice if the authors could discuss this further.

2) The current writing of the paper implicitly requires the reader to reference the supplement to understand several points.
I realize the paper is at the page limit, but It might be nice if some of these could be elaborated on in the main paper:
    1) Perhaps the definition of a Besov space should be given in the main paper rather than in the supplement, since a reader cannot really understand the paper without this. At very least, it should be noted in the main paper that the definition is given in the supplement.
    2) A brief description of how the DISTRSIM protocol works would be nice.
    3) In Section 4, a prior $\pi$ is mentioned but isn't described. In the supplement it looks like two different priors are used -- one uniform and one highly non-uniform. It would be nice if the authors could discuss the need for these two different priors, especially the non-uniform prior which isn't needed in the unconstrained setting.


-------AFTER READING AUTHOR REBUTTAL-------
I appreciate the points raised by other reviewers regarding the novelty and significance of the results, which rely heavily on prior work that is not very clearly explained (see my questions above). Since the authors did not answer my questions at all in the rebuttal, I have lowered my score from 8 to 7.

**Time Spent Reviewing:**

4

---

> ### Author Response · Authors · 2021-08-10
> **We thank the reviewer...**
>
> ... for their time and suggestions. We will take their comments into account when updating the paper and include the corresponding details in the main body.

---

> ### Author Response · Authors · 2021-09-13
> **Upon reading the updated review...**
>
> ... we apologize for not having responded point by point to the reviewer's suggestions. However, as mentioned in our response, we read
> the reviewer's suggestions, and agree -- and (as stated) will take them into account and implement said suggestions. As there were no
> questions (only suggestions), we didn't feel the need to explicitly say so for each of them.

---

### Official Review · Reviewer_83Tb · 2021-07-16

**Rating:** 7
**Confidence:** 3

**Summary:**

The paper studies a quantization approach for nonparametric density estimation in the Besov space with communication constraints.

**Limitations And Societal Impact:**

See my main review.

**Main Review:**

The proposed algorithm is the following: take the empirical coefficients and 1) if the number of coefficients (at the minimax wavelet cutoff level) is < communication constraint, suffering no quantization loss, and 2) if the number of coefficients is > communication constraint, use a simulate-and-infer approach to quantize the number of coefficients down to communication constraint.

I think the approach is reasonable and the theoretical analysis shows that the statistical rates are minimax optimal. I recommend acceptance.

I only have one comment: wavelet based approaches, similar to other methods that project to a family of basis functions, are convenient for theoretical analysis but usually not very practical because in practical the data are usually multi-variate and it is somewhat cumbersome to construct multivariate basis functions. Is it possible to design a communication efficient algorithm for some more practical methods, such as kernel smoothing or local polynomial regression?

**Time Spent Reviewing:**

0.5

---

> ### Author Response · Authors · 2021-08-10
> **We would like to thank...**
>
> ... the reviewer for their time and comments. We haven't considered the two methods mentioned, but will give it some thought.

---

### Official Review · Reviewer_KsRx · 2021-07-22

**Rating:** 6
**Confidence:** 3

**Summary:**

The paper addresses learning densities in Bezov space under communication constraints, primarily in a non-interactive setting. In fact, Theorems 1.1 and 1.3 together suggest that interaction may not help much in achieving a better performance. The proposed algorithms leverage wavelet-induced sparsity, vector quantization, and distributed (sample) simulation, but not the standard Fourier basis.

**Limitations And Societal Impact:**

The results are mainly theoretical. I think the limitations are clear from the assumptions. I do not foresee any potential negative societal impact.

**Main Review:**

Strengths

- The paper reads well, though I feel that the presentation of the appendices looks better than that of the submission. But the organization of the appendices is similar to journal papers.
- The theoretical results seem to be a reasonable contribution to the topic. The upper and lower bounds almost match. The idea of leveraging wavelets' sparsity is also natural. Combining several techniques and showing the resulting algorithm works well seems nontrivial.
- The fact that the lower bound also applies to the interactive setting is a plus to the paper's merits.

Weaknesses

- It might be reasonable to provide a guide for the appendices. There are no explicit statements (in the main paper) about which section proves which result. For example, I wonder whether Theorem 3.3 on the simulation protocol of [3] is a new result or not.
- Lines 114 - 119 state that the analysis in the paper draws upon ideas from [7] but differs from the latter, as [7] does not use techniques like distributed simulation. It seems reasonable to mention here that distributed simulation is also not a new idea and appears in [3].
- Line 133 mentioned that the lower bounds rely on the framework in [2]. I wonder if the authors can illustrate, on a high level, how different the extensions are from the original.
- I am pretty interested in seeing how the proposed algorithms perform in practice. I suggest the authors conduct some experiments on synthetic or natural datasets.

Minor

- Line 289 is apparently under full. I suggest the authors adjust the equations accordingly.





**Time Spent Reviewing:**

5

---

> ### Author Response · Authors · 2021-08-10
> **Thank you for your time reviewing our paper; we respond to your comments below.**
>
> - "It might be reasonable to provide a guide for the appendices. There are no explicit statements (in the main paper) about which section proves which result. For example, I wonder whether Theorem 3.3 on the simulation protocol of [3] is a new result or not."
>
> We agree that the paper might be a bit hard to navigate, and will add a paragraph on organization at the end of Section 1. We will also add an organizational paragraph at the beginning of the appendices.
> Regarding the example raised: Theorem 3.3 is a result in [3], which we state in view of using it as a blackbox. We will add a more explicit reference in the theorem header.
>
> - "Lines 114 - 119 state that the analysis in the paper draws upon ideas from [7] but differs from the latter, as [7] does not use techniques like distributed simulation. It seems reasonable to mention here that distributed simulation is also not a new idea and appears in [3]."
>
> We do acknowledge this throughout our discussion, and never intended to claim the idea of distributed simulation as ours; we apologise if this was the impression given. In particular, we cite [3] in Lines 33, 81, 128, 209-217 (for instance, in l.215 (Theorem 3.3), we explicit write “the simulation protocol of [3], denoted DISTRSIM”).
>
> - "Line 133 mentioned that the lower bounds rely on the framework in [2]. I wonder if the authors can illustrate, on a high level, how different the extensions are from the original."
>
> Lines 304-305 outline the result we borrow from [2]. This is detailed in Section D.1 in the Appendix; we will add an explicit reference to this in the main body, and (if space allows), expand a little the discussion in lines 304-305.
>
> - "I am pretty interested in seeing how the proposed algorithms perform in practice. I suggest the authors conduct some experiments on synthetic or natural datasets."
> We refer to our answer to Reviewer 4wEN, who raised a similar point.
>
> - "Minor: Line 289 is apparently under full. I suggest the authors adjust the equations accordingly."
>
> We will modify the equations to address this.

---

### Decision · Program_Chairs · 2021-09-27

**Decision:**

Accept (Poster)

**Comment:**

The paper studies estimation of densities in Besov spaces in a distributed interactive setting with communication constraints.
The obtained estimator is nearly minimax optimal. This is a nice contribution to the literature on distributed adaptive density estimation.
I would encourage the authors to revise the paper to include the reviewers' comments into the final version of the manuscript.